# CLIENT-CENTRIC FEDERATED LEARNING UNDER DYNAMIC MIXTURES OF DISTRIBUTIONS

## ABSTRACT

Conventional federated learning (FL) frameworks follow a server-centric model where the server determines session initiation and client participation. We introduce Client-Centric Federated Learning (CCFL), a novel client-centric FL framework that puts clients as the driving role of FL sessions. In CCFL, each client independently and asynchronously updates its model by uploading a locally trained model to the server and receiving a customized model tailored to its local task. The server maintains a repository of cluster models, iteratively refining them using received client models. Our framework accommodates complex dynamics in clients' data distributions, characterized by time-varying mixtures of cluster distributions, enabling rapid adaptation to new tasks with high performance. We propose novel strategies for accurate server estimation of clients' data distributions. CCFL offers clients complete autonomy for model updates, enhances model accuracy, and significantly reduces client computation, communication, and waiting time. We provide a theoretical analysis of CCFL's convergence. Extensive experiments across various datasets and system settings highlight CCFL's substantial advantages in model performance and computation efficiency over baselines.

## 1 INTRODUCTION

Federated Learning (FL) (McMahan et al., 2017) is a distributed learning framework that allows for collaborative training of a global model across multiple clients while keeping their raw data local. To tackle the problem of clients' non-iid data distributions, personalized FL (Tan et al., 2022) frameworks have emerged to offer each client a tailored model. However, in nearly all works within personalized FL, and even in the broader FL context, *the central locus of control invariably resides with the server.* That is, the server typically initiates training sessions and determines which clients should participate and when. Astonishingly, the following question has been conspicuously absent from the discourse: Why should a client always comply with the server's directives regarding model uploads? Are there not situations where network connectivity issues arise, or, indeed, a client simply does not want to share the model when server calls?

In this paper, we raise a brand-new FL framework: **C**lient-**C**entric Federated Learning (CCFL), *which empowers each individual client to assume a dominant role in the FL process.* In CCFL, each client device collects data from a mixture of distributions, whose mixing ratios may vary over time. Once a distribution shift is observed, the client may seek help from the server, who acts as a service provider, in updating its local model to match the new distribution. In real-life scenarios, this setting is commonplace. Consider a skincare maintenance application, where users' skin types exhibit complexity — perhaps featuring a combination of oiliness and dryness in different areas of skin, reflecting a mixture of distributions. Additionally, users' skin conditions may vary with seasons, leading to shifts in distributions. Another example is a retail chain with various branches, each of which sell commodities of different store categories. The commodities offered by these branches may evolve based on changing customer preferences, creating a dynamic mixture of various distributions. Note that in CCFL, each client possesses complete autonomy in deciding when to update its model, and the servers plays an assistive role for the clients to accomadating to their new distributions.

To tackle clients' data variations across multiple distributions, CCFL adopts the clustered FL setting where $K$ base cluster models are maintained at the server (Sattler et al., 2020a;b), which are used to update clients' models. In existing clustered FL works, a crucial consideration is to measure the data distributions of clients. Many works distribute all cluster models to clients, leaving it to

clients to determine the distribution based on local empirical loss (Ghosh et al., 2020; Mansour et al., 2020; Ruan and Joe-Wong, 2022). However, such an approach poses several challenges. Firstly, it places a significant communication burden to send all the cluster models. Secondly, it imposes substantial computational demands on clients, requiring them to calculate losses for each cluster and make comparisons. Some other approaches leverage distances between uploaded models to form client groups (Duan et al., 2021a), imposing impractical synchronization requirements on clients for data uploads. In sharp contrast, as illustrated in Figure 1, CCFL assigns the task of evaluating client data distribution to the server. Based on the model uploaded by a client, the server analyzes its data distribution, and updates the cluster models. Subsequently, the server generates a personalized model and sends it to the client. This significantly simplifying clients' communication and computation compared with previous clustered FL solutions.

In the context of above mentioned clustered FL, and building upon the client-centric foundation, we develop an asynchronous CCFL framework that focuses on maximizing clients' performance and minimizing clients' complexity. Notably, we introduce an effective newcomer cold start mechanism, a feature conspicuously absent in the majority of related works (Duan et al., 2021a; Zeng et al., 2023). Furthermore, our framework exhibits adaptability in addressing client distribution drift, a challenge specifically addressed in only one previous study (Duan et al., 2021b) within the context of clustered FL. *CCFL is the first clustered FL framework that focuses on client's autonomy, efficiency, and performance.* Compared to existing clustered FL works, client involvement remains minimal, as they only need to conduct local model training—a computationally modest task; users'

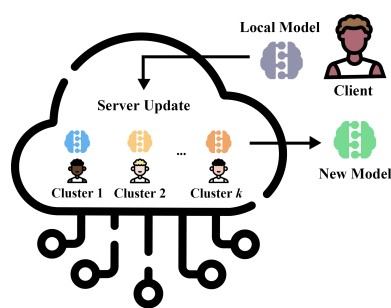

Figure 1: High-level view of CCFL.

communication overhead is equally reduced, with solely uploading and downloading *one single model*, and when to upload is left at their discretion. We provide convergence analysis that theoretically validates our framework. Extensive experiments over different datasets and network settings attest to the outstanding performance of CCFL. Notably, it significantly alleviates both communication and computational costs compared to existing works.

## 2 RELATED WORK

**Clustered Federated Learning (clustered FL)**. Hard clustering algorithms assume clients in the same group have identical data distribution (Briggs et al., 2020; Ghosh et al., 2020; Mansour et al., 2020); while soft clustering methods assume the data of each client follows a mixture of multiple distributions (Ruan and Joe-Wong, 2022; Li et al., 2021). In most cases, expectation-maximization (EM) methods are used to compute clients' distribution (Long et al., 2023; Ma et al., 2022; Ghosh et al., 2022), and global updates leverage methods based on FedAvg (Briggs et al., 2020). Some works add proximal terms on clients' objectives for personalization (Tang et al., 2021).

**Asynchronous Federated Learning (asynchronous FL)**. Asynchronous FL operates on resource-constrained devices (Xu et al., 2021). In typical asynchronous setups, the central server conducts global aggregation immediately upon receiving a local model (Xie et al., 2019; Wang et al., 2022; Chen et al., 2020), or a set of local models (Nguyen et al., 2022; Wu et al., 2020). These asynchronous clients may be grouped into tiers for updating based on factors like staleness or model similarities (Park et al., 2021; Wang and Wang, 2022), referred to as semi-asynchronous. However, this clustering typically contributes to a single global model, and sometimes, the server still selects the clients (Zhang et al., 2021). Existing clustered FL frameworks primarily operate within a synchronous setting. In the context of asynchronous FL, clients are sometimes grouped only to control staleness. Our framework is the first, to the best of our knowledge, to integrate clustered FL within an asynchronous setting.

**User-centric FL frameworks**. Few works have studied FL from a comprehensive user's perspective. Mestoukirdi et al. (2021; 2023) claim to be user-centric, but are indeed personalized FL frameworks dealing with communication burdens. In Khan et al. (2023), the authors point out that existing FL works take away clients' autonomy to make decisions themselves, and propose a token-based incentive mechanism that rewards personalized training. However, this work fails to consider the asynchrony among clients, making it insufficient to provide full autonomy to clients. Note that the shift in clients' distribution is distinct from Federated Continual Learning (FCL)(Yoon et al., 2021),

which primarily aims to minimize catastrophic forgetting. Our focus lies solely in enabling clients to seamlessly adapt their models to new data during distribution shifts.

# 3 PROBLEM DEFINITION

Consider an FL system with one central server and many distributed clients. The server maintains $K$ cluster models, each with a validation dataset $D_k$ corresponding to different distributions $P_1, \ldots, P_K$. The value of $K$ is determined a priori, according to the type of service (e.g., genders or ethnicities in the skincare service), or is deducted from a small amount of validation data collected in advance at the server. Given a loss function $l(w; x, y)$, each cluster $k \in [K]$ aims to find an optimal model $\mathbf{w}_k$ that minimizes the objective

$$F_k(\mathbf{w}_k) = \mathbb{E}_{(x,y) \sim P_k}[l(\mathbf{w}_k; x, y)]. \tag{1}$$

The training takes $T$ global epochs. For each epoch $t \in [T]$, some client $m$ collects local data following a mixture of distribution $P_m^t = \sum_{k=1}^{K} \mu_{mk}^t P_k$, with $\mu_{mk}^t \in [0, 1]$ and $\sum_{k=1}^{K} \mu_{mk}^t = 1$. Here $\mu_{mk}^t$ is the importance weight of cluster $k$ to client $m$ at epoch $t$. The importance weights may vary over time, and are unknown to the client. Each time when client $m$'s data distribution shifts, it may choose to fit the local model $w_m^t$ to the new distribution, by optimizing the local objective

$$h_m^t(w_m^t; \mathbf{w}_m^\tau) \triangleq \frac{1}{m_t} \mathbb{E}_{(x^i, y^i) \sim P_m^t} \sum_{i=1}^{m_t} l(w_m^t; x^i, y^i) + \frac{\rho}{2} \left\| w_m^t - \mathbf{w}_m^\tau \right\|^2. \tag{2}$$

Here $m_t$ is the number of data samples; $\rho$ is some scaling parameter; $\tau < t$ is the last epoch when client $m$ uploads its model $w_m^\tau$ to the server, and the server returns a model $\mathbf{w}_m^\tau$.

# 4 CLIENT-CENTRIC FEDERATED LEARNING

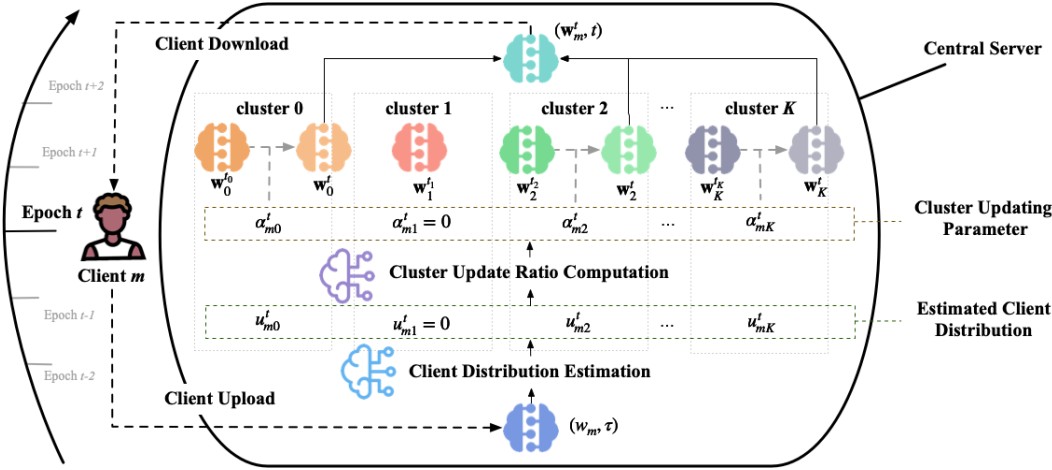

Figure 2: CCFL workflow. Client $m$ uploads model and timestamp tuple $(w_m, \tau)$ to the server. Server labels it at epoch $t$. In this figure, server estimates little distribution of $P_1$, and would not update cluster 1. An aggregated model based on client's estimated distribution is sent back after update.

## 4.1 CLIENT UPDATE

The user-centric architecture of CCFL empowers users to initiate the uploading process autonomously. To begin, client $m$ receives an initialization tuple from the server, comprising the global model and a timestamp, denoted as $(\mathbf{w}, t)$. Subsequently, the user adapts the global model $\mathbf{w}$ to its own dataset to obtain a personalized model $w_m$. After initialization, client $m$ retains the discretion to select when to upload the tuple of their local model and timestamp $(w_m, t)$, and then awaits the server's response, which serves to enhance their local performance.

**Client Data Shifts.** We assume the distribution shifts of clients between different epochs, i.e. for client $m$, it is possible that $\mu_{mk}^t \neq \mu_{mk}^{t'}$ for all $t \neq t', t, t' \in [T]$.

**Training and Uploading.** In order to establish a mutually beneficial system, clients are required to perform local training prior to model uploading (refer to Algorithm 2). The decision of when to upload rests entirely with the clients themselves. Furthermore, clients are advised to do *training and uploading* when there are shifts in data distribution to better align with the new data stream; or when a substantial amount of time has elapsed since the last upload to ensure synchronization with the server's state. Through this preliminary training session before uploading, the server gains valuable insights from the clients, facilitating the performance of cluster models.

**Scalable Client Integration.** We do not presuppose a fixed total number of clients. Our system is designed to be fully open and dynamic. A new user simply fetches an initialization tuple from the server, and starts the training and uploading process, seamlessly integrating into the system.

## 4.2 Server Update

---

**Algorithm 1:** DistributionEstimation & UpdateRaTioCompute

---

**Function** `DistributionEstimation`$(w_m, \mathbf{w}_0, ..., \mathbf{w}_K, D_0, ..., D_K)$:

    **foreach** $k \in [K]$ **do**

        $l_k \leftarrow F(w_m^t; D_k); d_{1k} \leftarrow \left\| F(\mathbf{w}_k^{t_k}; D_k) - F(w_m^t; D_k) \right\|_1; d_{2k} \leftarrow \left\| w_m^t - \mathbf{w}_k^{t_k} \right\|_2$

        `/* ` $l_{bar}, d_{1bar}, d_{2bar}$ ` are hyperparameters to control the scale  */`

        $l_k \leftarrow l_k - l_{bar}, d_{1k} \leftarrow d_{1k} - d_{1bar}, d_{2k} \leftarrow d_{2k} - d_{2bar}$

        `/* hyperparameters ` $c_1, c_2, c_1 + c_2 \in [0,1], u_{mk}^t \in [0,1], \sum_k u_{mk}^t = 1$ ` */`

        $u_{mk}^t \leftarrow \frac{1}{K-1} \cdot \left( c_1 \cdot \frac{\sum_{i \neq k} l_i}{\sum_i l_i} + c_2 \cdot \frac{\sum_{i \neq k} d_{1i}}{\sum_i d_{1i}} + (1 - c_1 - c_2) \cdot \frac{\sum_{i \neq k} d_{2i}}{\sum_i d_{2i}} \right)$

    `/* ` $A > 0$ ` is the amplifier, helping magnify the difference of`
        `distribution estimation among clusters.                  */`

    $u_{m0}^t, ..., u_{mK}^t \leftarrow \text{softmax}(u_{m0}^t \cdot A, ..., u_{mK}^t \cdot A)$

    **return** $u_{m0}^t, ..., u_{mK}^t$

**Function** `UpdateRaTioCompute`$(u_{m0}^t, ..., u_{mK}^t, \alpha_0, \tau)$:

    **foreach** $k \in [K]$ **do**

        $\alpha_{10}, ..., \alpha_{1K} \leftarrow u_{m0}^t, ..., u_{mK}^t$

        `/* If distribution content is less than preset bar ` $\alpha_{1bar}$`,`
            `do not update the cluster.                         */`

        $\alpha_{1max} \leftarrow \max(\alpha_{1k})$. **if** $\alpha_{1k} < \alpha_{1bar}$ **then** $\alpha_{1k} \leftarrow 0$; **else then** $\alpha_{1k} \leftarrow \alpha_{1k}/\alpha_{1max}$

        `/* ` $a, b$ ` are hyper-parameters to control staleness.     */`

        **if** $t_k - \tau < b$ **then** $\alpha_{2k} \leftarrow 1$; **else then** $\alpha_{2k} \leftarrow 1/(a(t_k - \tau) + 1)$

        `/* Hyper-parameter ` $\alpha_0$ ` governs the maximum extent of local`
            `model modification to the global cluster model.     */`

        $\alpha_{mk}^t \leftarrow \alpha_0 \cdot \alpha_{1k}\alpha_{2k}$     `/* ` $\alpha_{mk}^t \in [0, \alpha_0]$ ` */`

    **return** $\alpha_{m0}^t, ..., \alpha_{mK}^t$

---

Throughout the entire process of CCFL process, the server passively waits for the clients' uploads. Upon receipt of an upload, the server first updates and labels the client with global epoch $t$, then the server initiates a two-step evaluation process. Firstly, it checks if the client is too stale, i.e., when client $m$ uploads $(w_m; \tau)$ at epoch $t$. If $t - \tau > \tau_0$ ($\tau_0$ is a preset staleness threshold), the server refrains from updating and instead transmits a personalized model aggregated by cluster models. Otherwise, the server proceeds to estimate client $m$'s data distribution. Subsequently, it updates each cluster using a cluster-specific updating parameter and dispatches the personalized model back to the client.

**Distribution Estimation.** For each cluster $k$, a small public dataset $D_k$ derived from $P_k$ is stored at the server to do the clients' distribution estimation. Upon client $m$ uploading $w_m$ at epoch $t$ (referred to as $w_m^t$ for clarity), the estimation of client $m$'s data distribution hinges on several components, including $w_m^t$, the latest models of clusters denoted as $\mathbf{w}_k^{t_k}$ ($k \in [K]$), where $t_k$ is the last epoch when cluster $k$ is updated, and the validation dataset $D_k$. For distribution $k$, this estimation involves two distinct considerations. First, it takes into account the loss incurred by $w_m^t$ on distribution $P_k$, which is quantified by the empirical loss on validation dataset $D_k$, i.e. $F(w_m^t; D_k) = \mathbb{E}_{(x,y) \sim D_k} l(w_m^t; x, y)$. If $F(w_m^t; D_k) < F(w_m^t; D_{k'})$, it signifies that client $m$'s distribution $P_m^t$ may have a higher composition of distribution $P_k$ compared to $P_{k'}$. Second, if client

$m$ is not too stale ($t - \tau < \tau_0$), it is likely to resemble the latest global cluster model. This similarity is discernible either through the loss difference between the latest cluster model and the client's model on validation data, denoted as $\left\| F(\mathbf{w}_k^{t_k}; D_k) - F(w_m^t; D_k) \right\|_1$, or through the model distance, such as the $l_2$-norm distance, $\left\| w_m^t - \mathbf{w}_k^{t_k} \right\|_2$. Smaller values of these metrics signify a higher degree of similarity. Drawing from these observations, we employ Algorithm 1 to calculate the distribution estimation $u_{mk}^t, k \in [K]$. Based on the analysis presented in Section 5.2, we can reasonably posit that $u_{m0}^t, \ldots, u_{mK}^t$ serve as accurate estimations of the true importance weights $\mu_{m0}^t, \ldots, \mu_{mK}^t$. It's important to note that due to the potential distribution shifts on the client side, the server must recompute these weights every time a client initiates an upload.

**Clusters Updating.** The server updates the model of each cluster $k$ as follows

$$\mathbf{w}_k^t = (1 - \alpha_{mk}^t)\mathbf{w}_k^{t_k} + \alpha_{mk}^t w_m^t, \tag{3}$$

where $\alpha_{mk}^t$ is the updating ratio contributed by client $m$ to cluster $k$ at epoch $t$. The calculation of $\alpha_{mk}^t$ considers whether the client model predominantly originates from distribution $P_k$ (by the estimated proportion $u_{mk}^t$), and whether the client model is too stale (by $t_k$ and the timestamp $\tau$ to assess the degree of staleness). Detailed procedures for computing the updating ratio are elucidated in Algorithm 1. Note that only clusters with a non-zero updating rate ($\alpha_{mk}^t > 0$) undergo updates facilitated by client $m$'s model $w_m^t$.

**Aggregation and Feedback**. If client $m$ is not so stale ($t - \tau < \tau_0$), when all corresponding models finish updating, the server sends back the aggregated model $\mathbf{w}_m^t = \sum_{k=1}^K u_{mk}^t \mathbf{w}_k^{t_k}$ to client $m$. Otherwise, the new distribution would not be measured, and the server only sends back model $\mathbf{w}_m^t = \sum_{k=1}^K u_{mk}^\tau \mathbf{w}_k^{t_k}$ based on the measures at the last upload epoch $\tau$.

---

**Algorithm 2:** `CCFL`

---

**Input:** Server pre-trained model $\mathbf{w}_k^0$, server validation dataset $D_k \sim P_k$ ($k \in [K]$), staleness threshold $\tau_0 < T$, server update shreshold $\alpha_0 \in (0, 1)$
**Output:** Local model parameter $w_m$, global model parameter $\mathbf{w}_k$
**Initialization**: Server sends $(\mathbf{w}^0, 0)$ to each client, $\mathbf{w}^0 = \frac{1}{K}\sum_{k=1}^K \mathbf{w}_k^0$. Global epoch $t \leftarrow 0$.
  Run `Client()` thread and `Server()` thread asynchronously in parallel.
**Thread** `Server()`:
    **foreach** $k \in [K]$ **do** $t_k \leftarrow 0$. **while** $t \leq T$ **do**
      **while** *no client uploads* **do**
        /* Server passively waits for upload from clients. */
        Wait for client update. **if** *client m uploads* $(w_m, \tau)$ **then**
          $t \leftarrow t + 1$; $\mathbf{w}_m^t \leftarrow$ `ServerUpdate`$(w_m, \tau, t)$; send $(\mathbf{w}_m^t, t)$ to client $m$.

**Thread** `Client()`:
    **foreach** *client m in parallel* **do**
      Receive pair $(\mathbf{w}_m, 0)$ from server. set local model $w_m \leftarrow \mathbf{w}_m$, local timestamp $t_m \leftarrow 0$.
      **while** *active* **do**
        **if** *choose to upload* **then**
          Define $h_m(w_m; \mathbf{w}) = f_m(w_m; D_m) + \frac{\rho}{2}\|w_m - \mathbf{w}_m\|^2$
          **foreach** *local iteration h* **do**
            $w_{m,h} \leftarrow w_{m,h-1} - \gamma \nabla h_m(w_{m,h-1}; \mathbf{w}_m)$ /* learning rate $\gamma$ */
          Upload $(w_m, t_m)$ and wait for server response $(\mathbf{w}_m, t)$; $t_m \leftarrow t$

**Function** `ServerUpdate`$(w_m, \tau, t)$:
    /* If client deprecated, do not update global model. */
    **if** $t - \tau > \tau_0$ **return** $\mathbf{w}_m^t = \sum_{k=1}^K u_{mk}^\tau \mathbf{w}_k^{t_k}$.
    $u_{m0}^t, \ldots, u_{mK}^t \leftarrow$ `DistributionEstimate`$(w_m, \mathbf{w}_0, \ldots, \mathbf{w}_K, D_0, \ldots, D_K)$
    $\alpha_{m0}^t, \ldots, \alpha_{mK}^t \leftarrow$ `UpdateRatioCompute`$(u_{m0}^t, \ldots, u_{mK}^t, \alpha_0, \tau)$
    **foreach** $k \in [K]$ **do if** $\alpha_{mk}^t > 0$ **then** $\mathbf{w}_k^{t_k} \leftarrow (1 - \alpha_{mk}^t)\mathbf{w}_k^{t_k} + \alpha_{mk}^t w_m$, $t_k \leftarrow t$
    **return** $\mathbf{w}_m^t = \sum_{k=1}^K u_{mk}^t \mathbf{w}_k^{t_k}$.

---

The entire workflow of `CCFL` is depicted in Figure 2 and described in Algorithm 2.

### 4.3 CONVERGENCE ANALYSIS

We make some universal assumptions to assist the convergence analysis of CCFL.

**Assumption 1.** $F_k$ is $L_k$-smooth and $\mu_k$-strongly convex and for some $L_k, \mu_k > 0$ for all $k \in [K]$.

**Assumption 2.** Each client executes at least $H_{min}$ and at most $H_{max}$ local updates before updating.

**Assumption 3.** Denote $h_m^t(w; \mathbf{w}) = f(w) + \frac{\rho}{2} \|w - \mathbf{w}\|^2$, where $w, \mathbf{w} \in \mathbb{R}^d$ are respectively local and global models, we assume $\forall m, \forall t \in T$, we have $\|\nabla f_m^t(w)\|^2 \leq V_1$ and $\|\nabla h_m^t(w; \mathbf{w})\|^2 \leq V_2$.

**Assumption 4.** The distance of different clusters are bounded by $a_0\Delta \leq \|\mathbf{w}_k^* - \mathbf{w}_{k'}^*\| \leq \Delta$ for all $k \neq k', k, k' \in [K]$, where $\Delta \geq 0, 0 \leq a_0 \leq 1$ and $\mathbf{w}_k^* := \arg\min_{\mathbf{w}_k} F_k(\mathbf{w}_k)$.

**Assumption 5.** We assume there is always an upper bound on the $l_2$-norm of cluster $k$'s model $\mathbf{w}_k$, i.e, $\forall k \in [K], \|\mathbf{w}_k\| \leq a_k\Delta, a_k > 0$.

**Theorem 1.** *With above assumptions, for a small constant $\epsilon > 0$, assume we choose*
$\rho \geq \frac{2V_1 + \frac{1}{2}\|w - \mathbf{w}\|^2 + \sqrt{4\|w - \mathbf{w}\|^2(1 + V_1)\epsilon}}{2\|w - \mathbf{w}\|^2}$ *for all possible $w, \mathbf{w}$ in global and local iterations, then if cluster $k$ undergoes $S_k$ updates, Algorithm 2 would converges to:* $\mathbb{E}[\|\nabla F_k(w)\|^2] \leq$

$$\frac{\mathbb{E}[F_k(\mathbf{w}_0) - F_k(\mathbf{w}_{S_k})]}{\alpha_0 \gamma \epsilon S_k H_{min}} + \frac{\left(\frac{L_k}{2} + \rho H_{max} + \frac{\rho H_{max}^2}{2}\right)\gamma H_{max} V_2}{\epsilon H_{min}} + \frac{\sqrt{V_1}\left(2\sum_{i=1}^{K} a_i + (2K+1)a_k + K\right)\Delta}{\gamma \epsilon H_{min}} +$$
$$\frac{\left(\frac{L_k}{2} + \rho\right)\left(2\sum_{i=1}^{K} a_i + (2K+1)a_k + K\right)^2 \Delta^2}{\gamma \epsilon H_{min}}$$

*Discussions.* The theorem indicates that if a client's model $w$ undergoes continuous training on data from distribution $k$, meaning that a portion of the client's data consistently originates from distribution $k$, then the $l_2$-norm of the gradient of the model loss on cluster $k$ will converge to a specific point (always less than $\infty$). For any data distribution $k$ continuously sampled by a client, the proposed algorithm guarantees the client's model to have good performance on this particular distribution $k$.

## 5 EXPERIMENTS

### 5.1 SETUP

We create FL clustered datasets via three commonly used public datasets: FashionMNIST (Xiao et al., 2017), CIFAR-100 (Krizhevsky et al., 2009), MiniImageNet-100 (Vinyals et al., 2016). In order to simulate different distributions, we augment the datasets using rotation, and create the Rotated FashionMNIST, Rotated CIFAR-100 and Rotated MiniImagenet-100 datasets. Each dataset is applied by $i * \frac{360}{K}(i = 0, ..., K - 1)$ degrees of rotation to the images, resulting in $K$ clusters. In our experiment, we try $K = 2, 3, 4, 6$ to simulate an FL setup with clear cluster structure. **Rotated FashionMNIST**:Each rotated cluster has 60,000 training images and 10,000 testing images containing 10 classes. **Rotated CIFAR-100**: Each rotated cluster has 50,000 training images and 10,000 testing images containing 100 classes. **Rotated MiniImagenet-100**: Each rotated cluster has 48,000 training images and 12,000 testing images containing 100 classes. 2,000 images of testing images from each cluster of each dataset are used to pre-train the cluster models. Training models structures are listed in Appendix A.1 All the experiments are conducted using PyTorch version 1.9 on a single machine equipped with two Intel Xeon 6226R CPUs, 384GB of memory, and four NVIDIA 3090 GPUs. We compare our CCFL method with below baseline methods:

- FedSoft-Async. An asynchronous adaptation of the soft-clustering baseline Ruan and Joe-Wong (2022) is employed. Clients receive all global models from the server, and distribution is assessed by identifying the model with the smallest loss for each data point. Distribution values $\mu_{m0}, \ldots, \mu_{mK}$ are transmitted to the server alongside the local model for global updates. The clusters' update ratio, denoted as $\alpha_{mk}^t$, integrates the locally computed distribution $\mu_{mk}$ and staleness, given by $\alpha_{mk}^t := \alpha_0 \cdot \mu_{mk}\alpha_{2k}$, with $\alpha_{2k}$ computed in a similar manner as in CCFL. As there are no existing works addressing both asynchrony and soft-clustering concurrently in FL, FedSoft-Async serves as the most suitable baseline method.

- Local. The clients only do local optimizations and never upload the local models.

In the initialization phase, clients perform computations using the averaged cluster model. Each client possesses a dataset ranging from 500 to 2000 data points, with 40% to 90% originating from a primary distribution and the remainder from other cluster distributions. Upon completing the initialization, clients autonomously decide when to upload their models. After uploading, an accuracy evaluation is conducted initially on a test set matching the client's data distribution. Subsequently, upon receiving

the updated model from the server, a second accuracy evaluation is conducted to compare the local and global model improvements. Each upload-download cycle prompts clients to receive new data, necessitating recalculations and interactions with the server for updates. In the experiments presented in Table 1, the number of clients is 20 times the number of models. The value of global staleness control $\tau_0$ equals to the number of clients. In the FashionMNIST experiments, on average, each client undergoes 25 upload-download cycles, while in the CIFAR-100/MiniImagenet-100 experiments, each client averages 20 cycles. We set cluster aggregation parameter $\alpha_0 = 0.025, a = 10, b = 5$, client personalization parameter $\rho = 0.1$. Other parameters and explanations are left at Appendix A.1

## 5.2 BEHAVIOR OF CLUSTERS AND CLIENTS

Table 1: Client and Cluster Accuracy of FashionMNIST, CIFAR100, and MiniImagenet-100. Client accuracy is represented as the average accuracy, along with the standard deviation, of all clients at the final upload-download cycle. Cluster accuracy is depicted as the average accuracy, along with the standard deviation, of all clusters across all cycles averagely across repeated experiments. "Cli Brf" denotes the accuracy of the local model before uploading; "Cli Aft" represents the accuracy of the model received by the client. Average accuracy over 3 trials are reported.

| Dataset (cluster No.) | CCFL ACC. | | | FedSoft-Async ACC. | | | Local |
| | Cli Bfr | Cli Aft | Cluster | Cli Bfr | Cli Aft | Cluster | Client ACC. |
|---|---|---|---|---|---|---|---|
| FashionMNIST (2) | $.799\pm.011$ | $.836\pm.003$ | $.840\pm.008$ | $.798\pm.012$ | $.836\pm.003$ | $.833\pm.008$ | $.784\pm.014$ |
| FashionMNIST (3) | $.783\pm.015$ | $.822\pm.003$ | $.833\pm.003$ | $.780\pm.015$ | $.819\pm.004$ | $.822\pm.005$ | $.741\pm.057$ |
| FashionMNIST (4) | $.768\pm.020$ | $.801\pm.006$ | $.830\pm.005$ | $.763\pm.020$ | $.785\pm.006$ | $.795\pm.020$ | $.693\pm.076$ |
| FashionMNIST (6) | $.760\pm.021$ | $.779\pm.019$ | $.811\pm.009$ | $.753\pm.025$ | $.750\pm.024$ | $.740\pm.071$ | $.694\pm.072$ |
| CIFAR-100 (2) | $.373\pm.022$ | $.398\pm.006$ | $.423\pm.015$ | $.374\pm.026$ | $.404\pm.004$ | $.420\pm.001$ | $.279\pm.030$ |
| CIFAR-100 (3) | $.292\pm.037$ | $.313\pm.029$ | $.370\pm.031$ | $.281\pm.033$ | $.301\pm.008$ | $.354\pm.023$ | $.211\pm.033$ |
| CIFAR-100 (4) | $.354\pm.029$ | $.371\pm.012$ | $.427\pm.017$ | $.330\pm.037$ | $.355\pm.017$ | $.425\pm.022$ | $.259\pm.035$ |
| CIFAR-100 (6) | $.302\pm.032$ | $.319\pm.009$ | $.373\pm.024$ | $.278\pm.041$ | $.303\pm.016$ | $.382\pm.031$ | $.212\pm.035$ |
| MiniImagenet (2) | $.345\pm.022$ | $.372\pm.004$ | $.388\pm.009$ | $.346\pm.026$ | $.378\pm.003$ | $.393\pm.003$ | $.226\pm.032$ |
| MiniImagenet (3) | $.290\pm.030$ | $.311\pm.017$ | $.358\pm.010$ | $.275\pm.034$ | $.306\pm.011$ | $.352\pm.005$ | $.184\pm.029$ |
| MiniImagenet (4) | $.346\pm.025$ | $.371\pm.013$ | $.406\pm.007$ | $.323\pm.034$ | $.366\pm.008$ | $.403\pm.007$ | $.215\pm.028$ |
| MiniImagenet (6) | $.312\pm.028$ | $.336\pm.008$ | $.387\pm.012$ | $.283\pm.037$ | $.325\pm.011$ | $.383\pm.016$ | $.192\pm.027$ |

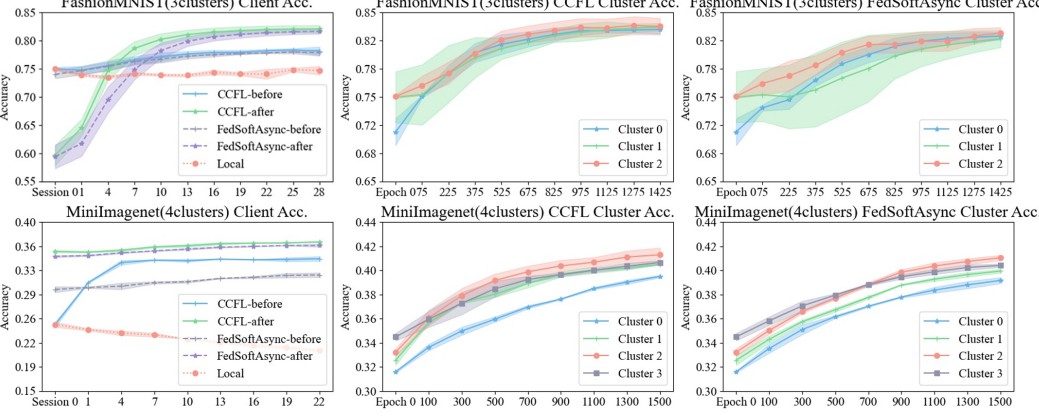

Figure 3: Accuracy of clients and clusters on FashionMNIST(3 clusters) and MiniImagenet (4 clusters). Average accuracy of clients is shown for equal times of upload-download cycles. Shaded areas represent variances across 3 trials.

**Accuracy Behavior.** Table 1 presents a comprehensive overview of client and cluster accuracy. Notably, both CCFL and FedSoft-Async exhibit significant enhancements in client performance compared to only local training, underscoring the importance of clients staying synchronized with the server. Across most experiments, CCFL outperforms FedSoft-Async for both clients and clusters, particularly when dealing with larger $K$. Figure 3 provides a performance analysis for a subset of experiments. Additional details can be found in Appendix A.3. In FashionMNIST experiments, both CCFL and FedSoft-Async require a few training epochs for the downloaded global model to surpass the performance of their locally uploaded counterparts. During this period,

clusters may experience a temporary dip in performance, and we refer to it as the "preparation period". This preparatory phase can be executed effectively through limited-scale $\alpha$-testing before software release. It's worth noting that this phenomenon is not observed in CIFAR-100 and MiniImagenet datasets due to their more complex prediction tasks, where the upload-download cycles with the server significantly aid clients in mitigating overfitting issues arising from limited data availability.

**Distribution Estimation.** To assess the accuracy of the distribution estimation outlined in Algorithm 1 in representing the true distribution, we conduct empirical comparisons between the estimation outcomes of `CCFL` and those obtained using `FedSoft-Async`. To quantify this assessment, we employ the KL-divergence metric, which measures the information loss when one distribution approximates another, denoted as $KL(P||Q) = \sum P(x) \log(P(x)/Q(x))$, where $P$ represents the true distribution, and $Q$ represents the estimated distribution. Lower KL divergence values signify more accurate estimation. The KL-divergence results for all the aforementioned experiments are depicted in Figure 4(b). We normalize the divergence rate of `FedSoft-Async` to 1 and record the proportional ratio of `CCFL`. Across all experiments, `CCFL` exhibits superior distribution estimation performance compared to `FedSoft-Async`, whose estimation method is commonly utilized in clustered FL works for distribution analysis.

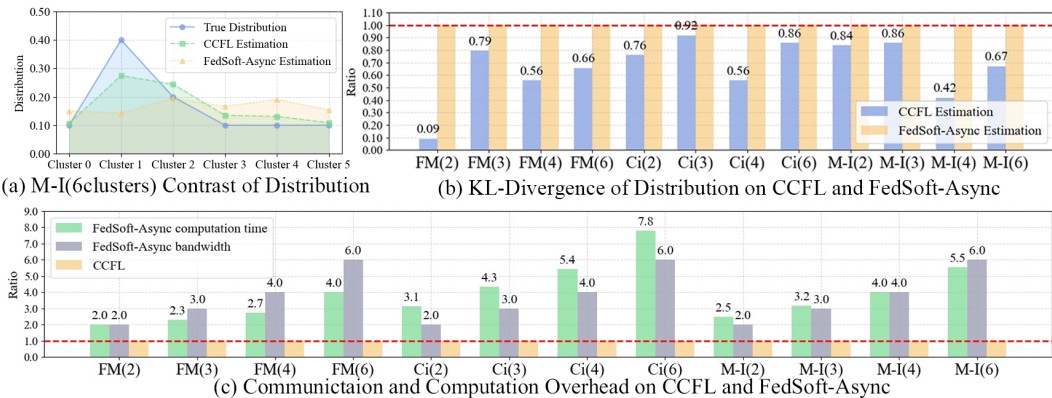

(a) M-I(6clusters) Contrast of Distribution
(b) KL-Divergence of Distribution on CCFL and FedSoft-Async
(c) Communicaiton and Computation Overhead on CCFL and FedSoft-Async

Figure 4: (a) is one distribution estimation made in MiniImagenet (6clusters) experiment; `CCFL` and `FedSoft-Async` results are contrasted to the true distribution. (b) is the KL-divergence of distribution estimation of `CCFL` and `FedSoft-Async` to the true distribution across all experiments. (c) is the communication and computation overhead contrast of `FedSoft-Async` with `CCFL`. FM($k$) denotes FashionMNIST ($k$ clusters), Ci as CIFAR-100, M-I as MiniImagenet-100.

**Communication and Computation Overhead.** We conduct a comparative analysis of the communication and computation overhead between `FedSoft-Async` and `CCFL`, as illustrated in Figure 4(c). Specifically, we focus on download sessions for communication overhead evaluation, as both methods upload one local model during upload sessions. We normalize both the communication and computation overhead of `CCFL` to 1, and record the proportional ratio of `FedSoft-Async`. Due to the fact that clients in `CCFL` solely download an aggregated global model and do not engage in additional computations beyond local model training, the communication and computation overhead is significantly reduced compared to `FedSoft-Async`. This highlights the lightweight and client-centric nature of our approach.

## 5.3 ABLATION STUDY

In order to comprehensively evaluate the robustness and performance of our framework, we conduct an ablation study on the FashionMNIST(4 clusters) and CIFAR100(4 clusters) datasets. The results of this study are depicted in Figure 5. **Multiple Clients:** We conduct experiments with varying numbers of clients of 100, 250, 500, 1000. Remarkably, the average accuracy of both clients and clusters exhibited minimal variation across different client counts. This observation underscores the robustness of our system. **Different $\rho$ Values:** We experiment with $\rho$ values set to 0.01, 0.1, 0.5, and 1. The results on both FashionMNIST and CIFAR100 datasets reveal that smaller $\rho$ values consistently lead to improved cluster accuracy. However, smaller $\rho$ values, as observed in CIFAR-100, result in suboptimal client local training performance before uploading, presenting a challenge. This can be attributed to similarities among cluster models, arising from generating clusters via various degrees of image rotation. These inherent similarities improve the aggregated data performance

across diverse distributions, consistent with Ruan and Joe-Wong (2022). Additionally, smaller $\rho$ values increase the risk of client overfitting to local data, further degrading local performance. **Global Adjustments:** To better regulate clients' contributions to global models, we introduce an adjustment technique in our experiments. During each client's update session, we record the values of $l_k$, $d_{1k}$, and $d_{2k}$ for each cluster $k$. Over time, this data accumulation create a reference database resembling normal distributions. Subsequently, after a certain number of epochs, the uploaded models undergo adjustments based on thresholds derived from the aforementioned database: if any of the uploaded model's $l_k$, $d_{1k}$, and $d_{2k}$ for given cluster $k$ exceeds 70% of the database, this client model is refused by the server to update global model $k$. This adjustment begins either after half of the training session, after 7/10 of the session, or not at all. Though accuracy does not changes, we believe this adjustment mechanism acts as a filter, potentially preventing certain clients' models from negatively impacting the server's model due to the non-iid nature of clients' data distribution. Ablation study with different size of public dataset on the server and data distribution without changes can be found in A.4. This section sheds light on the versatility and robustness of our CCFL framework, showcasing its adaptive ability to various scenarios and configurations while maintaining stable performance.

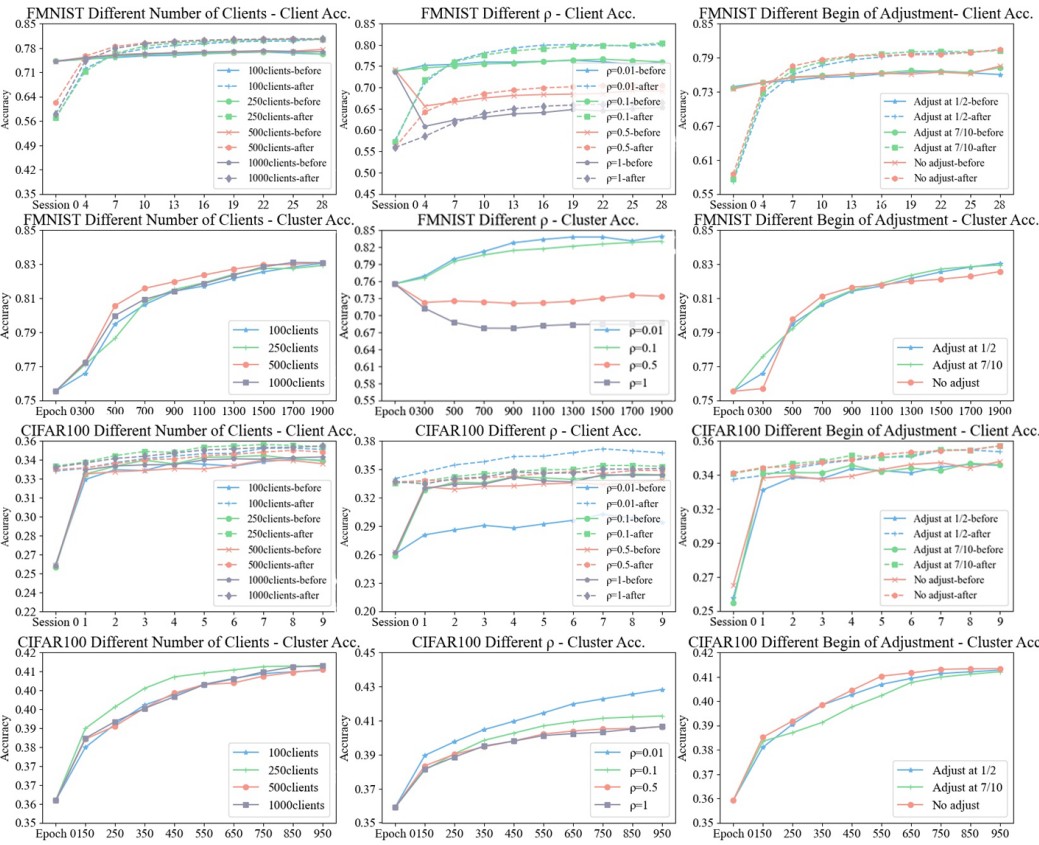

Figure 5: Ablation study on FashionMNIST (4 clusters) and CIFAR-100 (4 clusters). The clients undergo average 20 (FashionMNIST) / 10 (CIFAR-100) upload-download cycles in every experiment. Average accuracy of clients and clusters are recorded.

## 6 CONCLUSION

In summary, our paper introduces the Client-Centric Federated Learning (CCFL) framework, an approach that redefines the traditional server-centric FL paradigm. In this setting, clients independently decide when to upload their local models, resulting in rapid and personalized model updates from the server, who maintains multiple cluster models. Compared to existing clustered FL works, it significantly reduces computation and communication costs for clients. Moreover, CCFL accommodates dynamic clients' data distributions. Our experiments on FashionMNIST, CIFAR100 and MiniImagenet-100 datasets underscore CCFL's robustness and performance across different configurations. In conclusion, CCFL bridges the gap between user-centricity and model refinement, making it a pioneering framework in the FL landscape.

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

# A APPENDIX

## A.1 MODEL STRUCTURES, HYPER-PARAMETERS AND EXPLANATIONS.

We use CNN models to train FashionMNIST and ResNet18 to train Cifar-100 and MiniImagenet-100. The model structures are as follows.

- CNN model: This CNN model consists of two convolutional layers with 5x5 kernels and ReLU activation, each followed by max-pooling layers with 2x2 kernels. This is followed by two fully-connected layers with 512 and 10 neurons, respectively. The model takes 28x28 grayscale images as input and produces class probabilities for 10 classes. It employs ReLU activation throughout.

- ResNet18: ResNet-18 is characterized by its residual blocks, and it's a smaller variant of the original ResNet architecture. The model consists of two types of residual blocks: BasicBlock, containing two convolutional layers with 3x3 kernels and ReLU activations, along with batch normalization. It also includes shortcut connections to handle different input and output dimensions when the stride is not equal to 1; BottleNeck, including three convolutional layers with 1x1, 3x3, and 1x1 kernels, along with ReLU activations and batch normalization and uses shortcut connections for dimension matching. The ResNet-18 architecture is structured around an initial convolutional layer with 64 output channels and a 3x3 kernel with padding set to 1, followed by four stages of residual blocks. Each stage contains a specific number of residual blocks, with the configuration typically set as [2, 2, 2, 2].Within each residual block, the block type can be either BasicBlock or BottleNeck. The number of output channels in the convolutional layers is typically set according to the block type: 64 for BasicBlock and 256 for BottleNeck. Stride values are often set to 1 for BasicBlock and 2 for BottleNeck to achieve downsampling. The expansion factor is 1 for BasicBlock and 4 for BottleNeck. The model uses adaptive average pooling to produce a fixed-sized output, typically (1, 1) spatial dimensions, and a fully-connected layer for classification, with the number of output classes of 100 for CIFAR-100 and activation functions serving as essential hyperparameters.

For all experiments, batch size is chosen as 128. For CNN model, Adam optimizer is chosen with weight decay of 0.005 and learning rate of 0.01. For ResNet models, SGD opitimizer is chosen with weight decay of 5e-4, momentum of 0.9 and learning rate as 0.1. Cross entropy loss is computed for optimization through the experiments. Distribution estimation and server updating parameters are listed as follows in Table 2.

In the experiment, each client is initially assigned a main cluster index before training. During the training's outset, every client is randomly provided with 500-2000 data samples, with 40%-90% sourced from the designated main cluster and the remainder from other cluster distributions. Clients conduct local training and autonomously decide when to upload their models. If a client initiates an upload session, the accuracy of the uploaded model is first evaluated on the test sets (consisting of the same distribution as their training sets), termed as "Client Before Accuracy." Subsequently, upon downloading the new personalized model, the accuracy of the received model is assessed as "Client After Accuracy." After each update of every single client, he would receive new data sampled as the same way mentioned above. Since the distribution has changed, the client needs to do training on new data again and decides the next upload.

Table 2: Distribution estimation and server updating parameters. *min* means to use the minimum of the computed $l_k$ (or $d_{1k}, d_{2k}$) as bar. *ave* means to use the average of the computed $\alpha_{1k}$ as bar. If multiple values are resented in $A$, it means to do *softmax* multiple times with respective given value. Global adjustments described in section 5.3 starts from given percent of total epochs.

| Dataset (cluster No.) | Distribution Estimation | | | | | | Server Updating | Adjusting |
|---|---|---|---|---|---|---|---|---|
| | $c_1$ | $c_2$ | $A$ | $l_{bar}$ | $d_{1bar}$ | $d_{2bar}$ | $\alpha_{1bar}$ | % |
| FashionMNIST (2) | 0.5 | 0.4 | 3 | 8 | 0 | 0 | ave | 50 |
| FashionMNIST (3) | 0.5 | 0.25 | 3 | 8 | 0 | 0 | ave | 50 |
| FashionMNIST (4) | 0.5 | 0.25 | 7 | 8 | 0 | 0 | ave | 50 |
| FashionMNIST (6) | 0.7 | 0.2 | 15 | 8 | 0 | 0 | ave | 50 |
| CIFAR-100 (2) | 0.5 | 0.2 | 1 | 40 | 0 | 0 | ave | 50 |
| CIFAR-100 (3) | 0.5 | 0.2 | 10 | min | min | min | ave | 50 |
| CIFAR-100 (4) | 0.5 | 0.2 | 10 | min | min | min | ave | 50 |
| CIFAR-100 (6) | 0.5 | 0.2 | 10 | min | min | min | ave | 50 |
| MiniImagenet (2) | 0.5 | 0.2 | 1 | 70 | 0 | 0 | ave | 50 |
| MiniImagenet (3) | 0.5 | 0.4 | 7 | min | min | min | ave | 50 |
| MiniImagenet (4) | 0.6 | 0.3 | 15 | min | min | min | ave | 50 |
| MiniImagenet (6) | 0.6 | 0.3 | 10, 10 | min | min | min | ave | 50 |

## A.2 PROOF OF THEOREM 1.

Without the loss of generality, we assume client $m$ uploads $(w_m^t, \tau)$ to the server at epoch $t$. We assume the client is not stale ($t - \tau < \tau_0$), and cluster $k$'s model $\mathbf{w}_k^{t_k}$ would be updated with parameter $\alpha_{mk}^t > 0$. We assume $w_m^t$ is the result of applying $H_{min} \le H \le H_{max}$ local updates to $\mathbf{w}_m^\tau$. We define $H_k(w; \mathbf{w}) = F_k(w) + \frac{\rho}{2} \|w - \mathbf{w}\|^2$. For convenience we denote $w_{m,h}^t$ as $w_h$ ($h \in [H]$), $\mathbf{w}_m^\tau$ as $\mathbf{w}_\tau$.

Conditioning on $w_{h-1}$, for $\forall h \in [H]$ we have

$$\mathbb{E}[F_k(w_h) - F_k(\mathbf{w}^*)] \le \mathbb{E}[H_k(w_h; \mathbf{w}_\tau) - F_k(\mathbf{w}^*)] \le \mathbb{E}[H_k(w_h; \mathbf{w}_\tau)] - F_k(\mathbf{w}^*) \quad (4)$$

With the updating function

$$w_h = w_{h-1} - \gamma \nabla h_m(w_{h-1}; \mathbf{w}) \quad (5)$$

and Taloy's Expansion $f(w - y) = f(x) - \langle \nabla f(x), y \rangle + \frac{1}{2} \nabla^2 f(x) y^2$ and using $L_k$-smoothness,

$$
\begin{aligned}
&H_k(w_h; \mathbf{w}_\tau) \\
&= H_k(w_{h-1} - \gamma \nabla h_m(w_{h-1}; \mathbf{w}_\tau); \mathbf{w}_\tau) \\
&= H_k(w_{h-1}; \mathbf{w}_\tau) - \langle \nabla H_k(w_{h-1}; \mathbf{w}_\tau), \gamma \nabla h_m(w_{h-1}; \mathbf{w}_\tau) \rangle \\
&\quad + \frac{1}{2} \nabla^2 H_k(w_{h-1}; \mathbf{w}_\tau) \gamma^2 \|\nabla h_m(w_{h-1}; \mathbf{w}_\tau)\|^2 \\
&\le H_k(w_{h-1}; \mathbf{w}_\tau) - \langle \nabla H_k(w_{h-1}; \mathbf{w}_\tau), \gamma \nabla h_m(w_{h-1}; \mathbf{w}_\tau) \rangle + \frac{1}{2} L_k \gamma^2 \|\nabla h_m(w_{h-1}; \mathbf{w}_\tau)\|^2
\end{aligned}
\quad (6)
$$

Since $\|w_{h-1} - \mathbf{w}_\tau\|^2 \le H_{max}^2 \gamma^2 V_2$ and $\|h_m(w_{h-1}; \mathbf{w}_\tau)\|^2 \le V_2$, with equation 6,

$$\mathbb{E}[F_k(w_h) - F_k(\mathbf{w}^*)]$$
$$\leq H_k(w_{h-1}; \mathbf{w}_\tau) - F_k(\mathbf{w}^*) - \gamma \mathbb{E}[\langle \nabla H_k(w_{h-1}; \mathbf{w}_\tau), \nabla h_m(w_{h-1}; \mathbf{w}_\tau)\rangle]$$
$$+ \frac{1}{2} L_k \gamma^2 \mathbb{E}[\|\nabla h_m(w_{h-1}; \mathbf{w}_\tau)\|^2]$$
$$= F_k(w_{h-1}) - F_k(\mathbf{w}^*) + \frac{\rho}{2}\|w_{h-1} - \mathbf{w}_\tau\|^2 - \gamma \mathbb{E}[\langle \nabla H_k(w_{h-1}; \mathbf{w}_\tau), \nabla h_m(w_{h-1}; \mathbf{w}_\tau)\rangle]$$
$$+ \frac{1}{2} L_k \gamma^2 \mathbb{E}[\|\nabla h_m(w_{h-1}; \mathbf{w}_\tau)\|^2]$$
$$\leq F_k(w_{h-1}) - F_k(\mathbf{w}^*) - \gamma \mathbb{E}[\langle \nabla H_k(w_{h-1}; \mathbf{w}_\tau), \nabla h_m(w_{h-1}; \mathbf{w}_\tau)\rangle] + \frac{L_k \gamma^2}{2} V_2 + \frac{\rho H_{max}^2 \gamma^2}{2} V_2 \tag{7}$$

Take a small constant $\epsilon > 0$, and with inequality of arithmetic and geometric mean, if we choose some $\rho \geq \frac{2V_1 + \frac{1}{2}\|w_{h-1} - \mathbf{w}_\tau\|^2 + \sqrt{4\|w_{h-1} - \mathbf{w}_\tau\|^2(1 + V_1)\epsilon}}{2\|w_{h-1} - \mathbf{w}_\tau\|^2}$ for all possible $w_{h-1}, \mathbf{w}_\tau$, we have

$$\langle \nabla H_k(w_{h-1}; \mathbf{w}_\tau), \nabla h_m(w_{h-1}; \mathbf{w}_\tau)\rangle - \epsilon \|\nabla F_k(w_{h-1})\|^2$$
$$= \langle \nabla F_k(w_{h-1}) + \rho(w_{h-1} - \mathbf{w}_\tau), \nabla f_m(w_{h-1}) + \rho(w_{h-1} - \mathbf{w}_\tau)\rangle - \epsilon \|\nabla F_k(w_{h-1})\|^2$$
$$= \langle \nabla F_k(w_{h-1}), \nabla f_m(w_{h-1})\rangle + \rho\langle \nabla F_k(w_{h-1}) + \nabla f_m(w_{h-1}), w_{h-1} - \mathbf{w}_\tau\rangle$$
$$+ \rho^2 \|w_{h-1} - \mathbf{w}_\tau\|^2 - \epsilon \|\nabla F_k(w_{h-1})\|^2$$
$$\geq -\frac{1}{2}\|\nabla F_k(w_{h-1})\|^2 - \frac{1}{2}\|\nabla f_m(w_{h-1})\|^2 - \frac{\rho}{2}\|\nabla F_k(w_{h-1}) + f_m(w_{h-1})\|^2$$
$$- \frac{\rho}{2}\|w_{h-1} - \mathbf{w}_\tau\|^2 + \rho^2 \|w_{h-1} - \mathbf{w}_\tau\|^2 - \epsilon \|\nabla F_k(w_{h-1})\|^2$$
$$\geq -\frac{1}{2}\|\nabla F_k(w_{h-1})\|^2 - \frac{1}{2}\|\nabla f_m(w_{h-1})\|^2 - \rho\|\nabla F_k(w_{h-1})\|^2 - \rho\|\nabla f_m(w_{h-1})\|^2 \tag{8}$$
$$- \frac{\rho}{2}\|w_{h-1} - \mathbf{w}_\tau\|^2 + \rho^2 \|w_{h-1} - \mathbf{w}_\tau\|^2 - \epsilon \|\nabla F_k(w_{h-1})\|^2$$
$$= \|w_{h-1} - \mathbf{w}_\tau\|^2 \rho^2 - \left(\|\nabla F_k(w_{h-1})\|^2 + \|\nabla f_m(w_{h-1})\|^2 + \frac{1}{2}\|w_{h-1} - \mathbf{w}_\tau\|^2\right)\rho$$
$$- \left(\frac{1}{2}\|\nabla F_k(w_{h-1})\|^2 + \frac{1}{2}\|\nabla f_m(w_{h-1})\|^2 + \epsilon\|\nabla F_k(w_{h-1})\|^2\right)$$
$$\geq \|w_{h-1} - \mathbf{w}_\tau\|^2 \rho^2 - \left(2V_1 + \frac{1}{2}\|w_{h-1} - \mathbf{w}_\tau\|^2\right)\rho - (1 + V_1)\epsilon \geq 0$$

Thus, we have

$$\gamma\langle \nabla H_k(w_{h-1}; \mathbf{w}_\tau), \nabla h_m(w_{h-1}; \mathbf{w}_\tau)\rangle \geq \gamma\epsilon \|\nabla F_k(w_{h-1})\|^2 \tag{9}$$

Taking equation 9 into equation 7, we have

$$\mathbb{E}[F_k(w_h) - F_k(\mathbf{w}*)] \leq F_k(w_{h-1}) - F_k(\mathbf{w}*) - \gamma\epsilon \|\nabla F_k(w_{h-1})\|^2 + \frac{L_k \gamma^2}{2} V_2 + \frac{\rho H_{max}^2 \gamma^2}{2} V_2 \tag{10}$$

By iterating equation 10 for $h = 0, ..., H - 1$, we have

$$\mathbb{E}[F_k(w_h) - F_k(w_0)] \leq -\gamma\epsilon \sum_{h=0}^{H-1} \|\nabla F_k(w_h)\|^2 + \frac{H_{max} L_k \gamma^2}{2} V_2 + \frac{\rho H_{max}^3 \gamma^2}{2} V_2 \tag{11}$$

Since $w_0$ is initiated from $\mathbf{w}_\tau$, we can rewrite above equation as

$$\mathbb{E}[F_k(w_h) - F_k(\mathbf{w}_\tau)] \leq -\gamma\epsilon \sum_{h=0}^{H-1} \|\nabla F_k(w_h)\|^2 + \frac{H_{max}L\gamma^2}{2}V_2 + \frac{\rho H_{max}^3 V_2}{2}V_2 \qquad (12)$$

We know that cluster $k$ is not updated in every iteration $t$. If we relabel the iterations when cluster $k$ is updated as $0, 1, 2, ..., s-1, s = t$, then we have $\mathbf{w}_k^s = (1 - \alpha_{mk}^s)\mathbf{w}_k^{s-1} + \alpha_{mk}^s w_m^s$. For simplicity, we denote $\mathbf{w}_k^s$ as $\mathbf{w}_s$, $\alpha_{mk}^s$ as $\alpha_s$, then

$$\begin{aligned}
&\mathbb{E}[F_k(\mathbf{w}_s) - F_k(\mathbf{w}_{s-1})] \\
&\leq \mathbb{E}[H_k(\mathbf{w}_s; \mathbf{w}_{s-1}) - F_k(\mathbf{w}_{s-1})] \\
&= \mathbb{E}[H_k((1-\alpha_s)\mathbf{w}_{s-1} + \alpha_s w_m^s; \mathbf{w}_{s-1}) - F_k(\mathbf{w}_{s-1})] \\
&= \mathbb{E}[H_k((1-\alpha_s)\mathbf{w}_{s-1} + \alpha_s w_H; \mathbf{w}_{s-1}) - F_k(\mathbf{w}_{s-1})] \\
&\leq \mathbb{E}[(1-\alpha_s)H_k(\mathbf{w}_{s-1}; \mathbf{w}_{s-1}) + \alpha_s H_k(w_H; \mathbf{w}_{s-1}) - F_k(\mathbf{w}_{s-1})] \\
&= \mathbb{E}[\alpha_s(F_k(w_H) - F(\mathbf{w}_{s-1})) + \frac{\alpha_s\rho}{2}\|w_H - \mathbf{w_{s-1}}\|^2] \\
&\leq \alpha_s\mathbb{E}[F_k(w_H) - F_k(\mathbf{w}_{s-1})] + \alpha_s\rho\|w_H - \mathbf{w}_\tau\|^2 + \alpha_s\rho\|\mathbf{w}_\tau - \mathbf{w}_{s-1}\|^2
\end{aligned} \qquad (13)$$

We have

$$\begin{aligned}
&\|\mathbf{w}_\tau - \mathbf{w}_{s-1}\| \\
&= \|\mathbf{w}_m^\tau - \mathbf{w}_k^{s-1}\| \\
&= \left\|\sum_{i=1}^K u_{mi}^s \mathbf{w}_i^{t_{i,\tau}} - \mathbf{w}_k^{s-1}\right\| \\
&= \left\|\sum_{i=1}^K u_{mi}^s \mathbf{w}_i^{t_{i,\tau}} - \mathbf{w}_k^\tau + \mathbf{w}_k^\tau - \mathbf{w}_k^{s-1}\right\| \\
&= \left\|\sum_{i=1}^K u_{mi}^s \left(\mathbf{w}_i^{t_{i,\tau}} - \mathbf{w}_k^\tau\right) + \left(\mathbf{w}_k^\tau - \mathbf{w}_k^{s-1}\right)\right\| \\
&\leq \sum_{i=1}^K u_{mi}^s \left\|\mathbf{w}_i^{t_{i,\tau}} - \mathbf{w}_k^\tau\right\| + \left\|\mathbf{w}_k^\tau - \mathbf{w}_k^{s-1}\right\| \\
&\leq \sum_{i=1}^K \left\|\mathbf{w}_i^{t_{i,\tau}} - \mathbf{w}_k^\tau\right\| + \left\|\mathbf{w}_k^\tau - \mathbf{w}_k^{s-1}\right\|
\end{aligned} \qquad (14)$$

where $t_{i,\tau} \leq \tau$ is the last updating epoch before epoch $\tau$ for cluster $i \in [K]$. We can notice that

$$\begin{aligned}
&\left\|\mathbf{w}_i^{t_{i,\tau}} - \mathbf{w}_k^\tau\right\| \\
&= \left\|\mathbf{w}_i^{t_{i,\tau}} - \mathbf{w}_i^* + \mathbf{w}_i^* - \mathbf{w}_k^* + \mathbf{w}_k^* - \mathbf{w}_k^\tau\right\| \\
&\leq \left\|\mathbf{w}_i^{t_{i,\tau}} - \mathbf{w}_i^*\right\| + \|\mathbf{w}_i^* - \mathbf{w}_k^*\| + \|\mathbf{w}_k^* - \mathbf{w}_k^\tau\|
\end{aligned} \qquad (15)$$

Since $\|\mathbf{w}_k\| \leq a_k\Delta$, then $\|\mathbf{w}_k - \mathbf{w}_k^*\| \leq 2a_k\Delta$, thus

$$\left\|\mathbf{w}_i^{t_{i,\tau}} - \mathbf{w}_k^\tau\right\| \leq 2a_i\Delta + \Delta + 2a_k\Delta = (2a_i + 2a_k + 1)\Delta \qquad (16)$$

And,

$$\begin{aligned}
\left\|\mathbf{w}_k^\tau - \mathbf{w}_k^{s-1}\right\| &= \left\|\mathbf{w}_k^\tau - \mathbf{w}_k^* + \mathbf{w}_k^* - \mathbf{w}_k^{s-1}\right\| \\
&\leq \|\mathbf{w}_k^\tau - \mathbf{w}_k^*\| + \left\|\mathbf{w}^* - \mathbf{w}_k^{s-1}\right\| \leq 2a_k\Delta + 2a_k\Delta = 4a_k\Delta
\end{aligned} \qquad (17)$$

Thus,

$$\|\mathbf{w}_\tau - \mathbf{w}_{s-1}\| \leq \left( 2\sum_{i=1}^{K} a_i + (2K+1)a_k + K \right) \Delta \tag{18}$$

And with $\|w_H - \mathbf{w}_\tau\|^2 \leq H_{max}^2 \gamma^2 V_2$, from equation 13 we have

$$
\begin{aligned}
&\mathbb{E}[F_k(\mathbf{w}_s) - F_k(\mathbf{w}_{s-1})] \\
&\leq \alpha_s \mathbb{E}[F_k(w_H) - F_k(\mathbf{w}_{s-1})] + \alpha_s \rho H_{max}^2 \gamma^2 V_2 + \alpha_s \rho \left( 2\sum_{i=1}^{K} a_i + (2K+1)a_k + K \right)^2 \Delta^2 \\
&\leq \alpha_s \mathbb{E}[F_k(w_H) - F_k(\mathbf{w}_\tau) + F_k(\mathbf{w}_\tau) - F_k(\mathbf{w}_{s-1})] + \alpha_s \rho H_{max}^2 \gamma^2 V_2 \\
&\quad + \alpha_s \rho \left( 2\sum_{i=1}^{K} a_i + (2K+1)a_k + K \right)^2 \Delta^2
\end{aligned}
\tag{19}
$$

Using $L_k$-smoothness, we have

$$
\begin{aligned}
&\mathbb{E}[F_k(\mathbf{w}_\tau) - F_k(\mathbf{w}_{s-1})] \\
&\leq \langle \nabla F_k(\mathbf{w}_{s-1}), \mathbf{w}_\tau - \mathbf{w}_{s-1} \rangle + \frac{L_k}{2} \|\mathbf{w}_\tau - \mathbf{w}_{s-1}\|^2 \\
&\leq \|F_k(\mathbf{w}_{s-1})\| \|\mathbf{w}_\tau - \mathbf{w}_{s-1}\| + \frac{L_k}{2} \|\mathbf{w}_\tau - \mathbf{w}_{s-1}\|^2 \\
&\leq \sqrt{V_1} \left( 2\sum_{i=1}^{K} a_i + (2K+1)a_k + K \right) \Delta + \frac{L_k}{2} \left( 2\sum_{i=1}^{K} a_i + (2K+1)a_k + K \right)^2 \Delta^2
\end{aligned}
\tag{20}
$$

With the inequalities from equation 12, equation 20, we can rewrite equation 19 into

$$
\begin{aligned}
&\mathbb{E}[F_k(\mathbf{w}_s) - F_k(\mathbf{w}_{s-1})] \\
&\leq -\alpha_s \gamma \epsilon \sum_{h=0}^{H-1} \|\nabla F_k(w_h)\|^2 + \frac{H_{max} L_k \gamma^2}{2} \alpha_s V_2 + \frac{\rho H_{max}^3 \gamma^2}{2} \alpha_s V_2 \\
&\quad + \alpha_s \sqrt{V_1} \left( 2\sum_{i=1}^{K} a_i + (2K+1)a_k + K \right) \Delta + \frac{\alpha_s L}{2} \left( 2\sum_{i=1}^{K} a_i + (2K+1)a_k + K \right)^2 \Delta^2 \\
&\quad + \alpha_s \rho H_{max}^2 \gamma^2 V_2 + \alpha_s \rho \left( 2\sum_{i=1}^{K} a_i + (2K+1)a_k + K \right)^2 \Delta^2 \\
&= -\alpha_s \gamma \epsilon \sum_{h=0}^{H-1} \|\nabla F_k(w_h)\|^2 + \left( \frac{L_k}{2} + \rho H_{max} + \frac{\rho H_{max}^2}{2} \right) \gamma^2 H_{max} \alpha_s V_2 \\
&\quad + \alpha_s \sqrt{V_1} \left( 2\sum_{i=1}^{K} a_i + (2K+1)a_k + K \right) \Delta \\
&\quad + \alpha_s \left( \frac{L}{2} + \rho \right) \left( 2\sum_{i=1}^{K} a_i + (2K+1)a_k + K \right)^2 \Delta^2
\end{aligned}
\tag{21}
$$

Denoting $H_s$ as the number of the local iterations applied on client $m$ before be uploads at epoch $s$, by rearranging terms in equation 21, we have

$$\sum_{h=0}^{H_s-1} \|\nabla F_k(w_h)\|^2$$

$$\leq \frac{\mathbb{E}[F_k(\mathbf{w}_{s-1}) - F_k(\mathbf{w}_s)]}{\alpha_s \gamma \epsilon} + \frac{\left(\frac{L_k}{2} + \rho H_{max} + \frac{\rho H_{max}^2}{2}\right) \gamma H_{max} V_2}{\epsilon}$$

$$+ \frac{\sqrt{V_1} \left(2\sum_{i=1}^K a_i + (2K+1)a_k + K\right) \Delta}{\gamma \epsilon}$$

$$+ \frac{\left(\frac{L}{2} + \rho\right) \left(2\sum_{i=1}^K a_i + (2K+1)a_k + K\right)^2 \Delta^2}{\gamma \epsilon} \tag{22}$$

Denote $\tau_s$ as if client $m$ uploads at epoch $s$, the last time of client's communication (client last uploads at epoch $\tau_s$ before $s$), by taking total expectation, after $S_k$ global epoch on cluster $k$, where $S_k$ is the total number of validate updates on cluster $k$, we have

$$\mathbb{E}[\|\nabla F_k(w)\|^2]$$

$$= \mathbb{E}_{\tau_s, s \in \{0,\dots,T-1\}, h \in \{0,\dots,H_t'-1\}} \left[\|\nabla F(w_{\tau_s, h})\|^2\right]$$

$$= \frac{1}{\sum_{s=1}^{S_k} H_s} \sum_{s=1}^{S_k} \sum_{h=0}^{H_s-1} \|\nabla F_k(w_h)\|^2$$

$$\leq \frac{\mathbb{E}[F_k(\mathbf{w}_0) - F_k(\mathbf{w}_{S_k})]}{\alpha_0 \gamma \epsilon S_k H_{min}} + \frac{\left(\frac{L}{2} + \rho H_{max} + \frac{\rho H_{max}^2}{2}\right) \gamma H_{max} V_2}{\epsilon H_{min}}$$

$$+ \frac{\sqrt{V_1} \left(2\sum_{i=1}^K a_i + (2K+1)a_k + K\right) \Delta}{\gamma \epsilon H_{min}}$$

$$+ \frac{\left(\frac{L_k}{2} + \rho\right) \left(2\sum_{i=1}^K a_i + (2K+1)a_k + K\right)^2 \Delta^2}{\gamma \epsilon H_{min}} \tag{23}$$

### A.3 Other experiment Results.

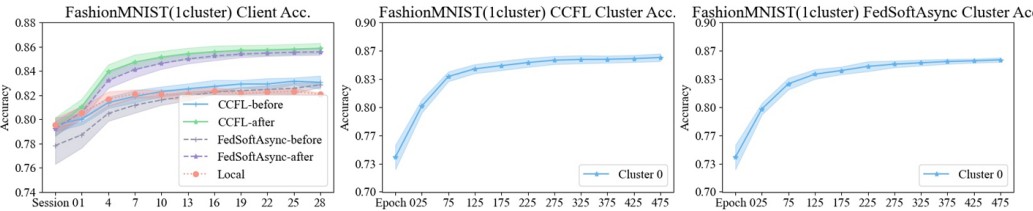

Figure 6: Timeflow accuracy of clients and clusters on FashionMNIST(1cluster). Average accuracy of clients is shown for equal times of upload-download cycles.

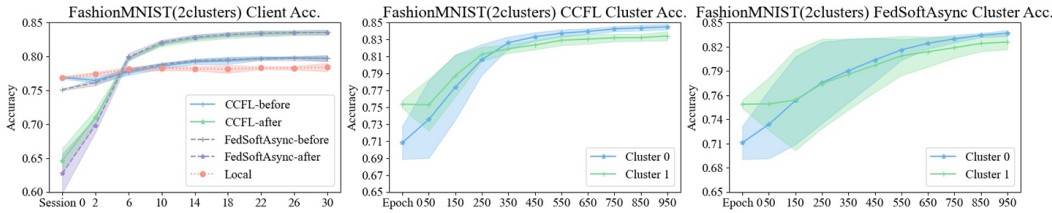

Figure 7: Timeflow accuracy of clients and clusters on FashionMNIST(2clusters). Average accuracy of clients is shown for equal times of upload-download cycles.

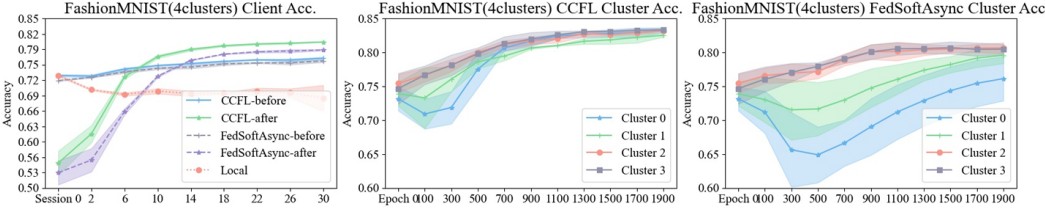

Figure 8: Timeflow accuracy of clients and clusters on FashionMNIST(4clusters). Average accuracy of clients is shown for equal times of upload-download cycles.

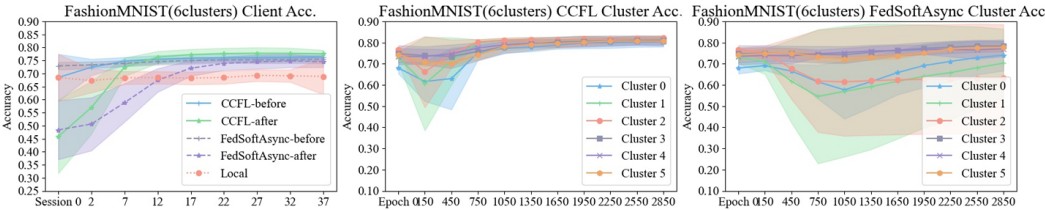

Figure 9: Timeflow accuracy of clients and clusters on FashionMNIST(6clusters). Average accuracy of clients is shown for equal times of upload-download cycles.

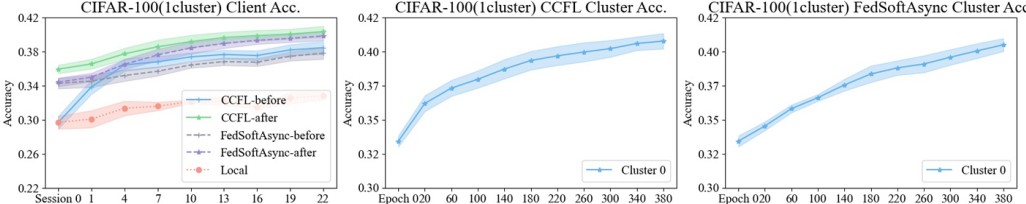

Figure 10: Timeflow accuracy of clients and clusters on CIFAR-100(1cluster). Average accuracy of clients is shown for equal times of upload-download cycles.

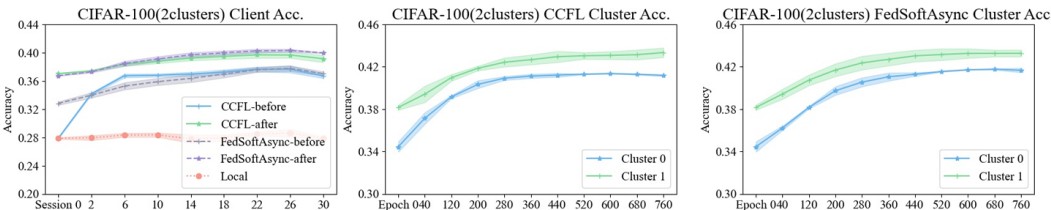

Figure 11: Timeflow accuracy of clients and cluseters on CIFAR-100(2clusters). Average accuracy of clients is shown for equal times of upload-download cycles.

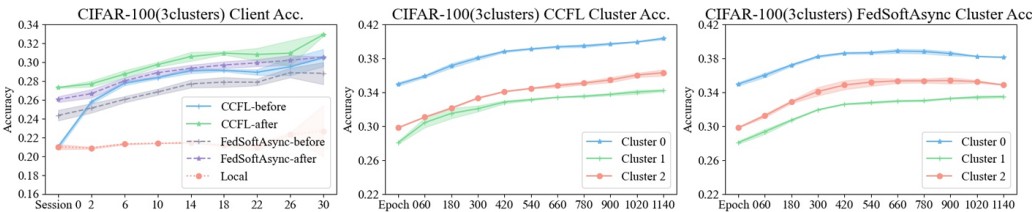

Figure 12: Timeflow accuracy of clients and clusters on CIFAR-100(3clusters). Average accuracy of clients is shown for equal times of upload-download cycles.

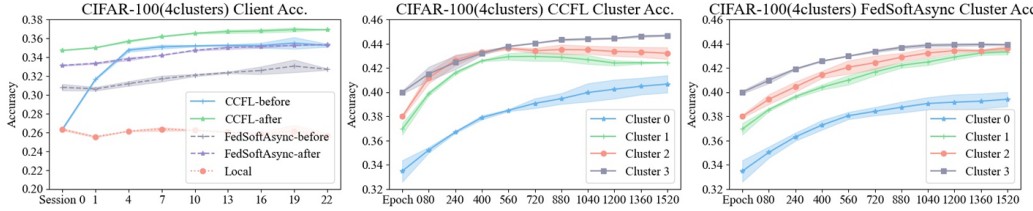

Figure 13: Timeflow accuracy of clients and clusters on CIFAR-100(4clusters). Average accuracy of clients is shown for equal times of upload-download cycles.

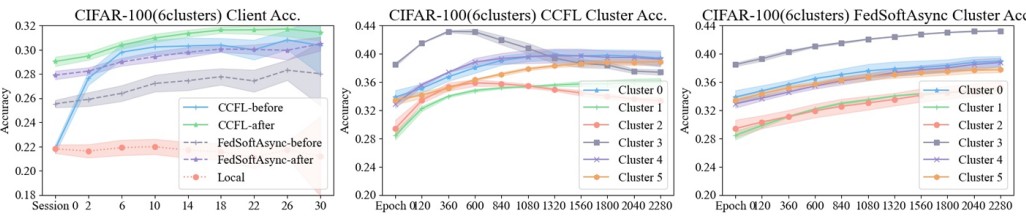

Figure 14: Timeflow accuracy of clients and clusters on CIFAR-100(6clusters). Average accuracy of clients is shown for equal times of upload-download cycles.

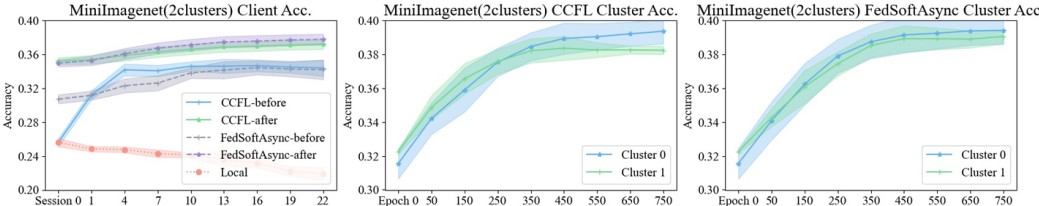

Figure 15: Timeflow accuracy of clients and clusters on MiniImagenet-100(2clusters). Average accuracy of clients is shown for equal times of upload-download cycles.

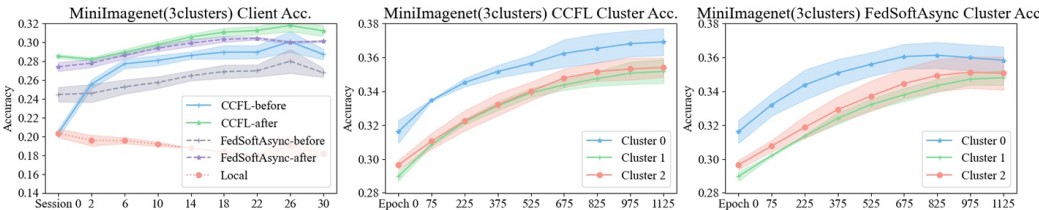

Figure 16: Timeflow accuracy of clients and clusters on MiniImagenet-100(3clusters). Average accuracy of clients is shown for equal times of upload-download cycles.

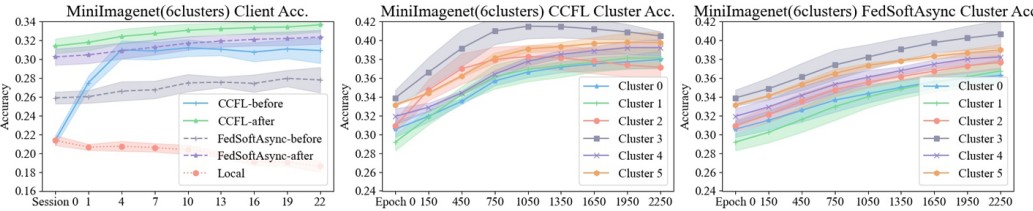

Figure 17: Timeflow accuracy of clients and clusters on MiniImagenet-100(6clusters). Average accuracy of clients is shown for equal times of upload-download cycles.

## A.4 MORE ABLATION STUDY.

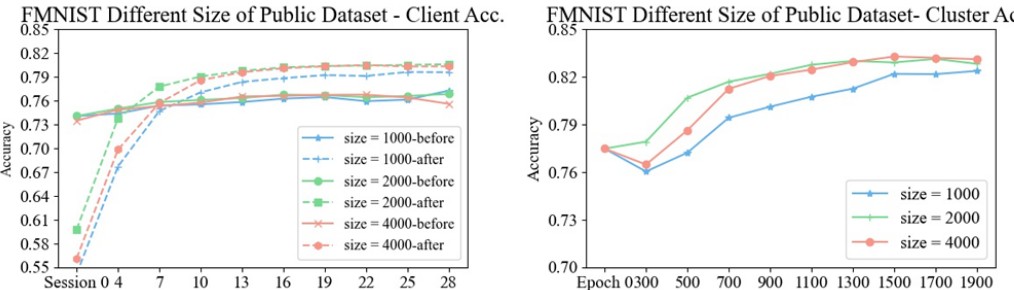

Figure 18: Timeflow accuracy of clients and clusters on FashionMNIST(4clusters) with different size of public dataset on the server. Average accuracy of clients is shown for equal times of upload-download cycles.

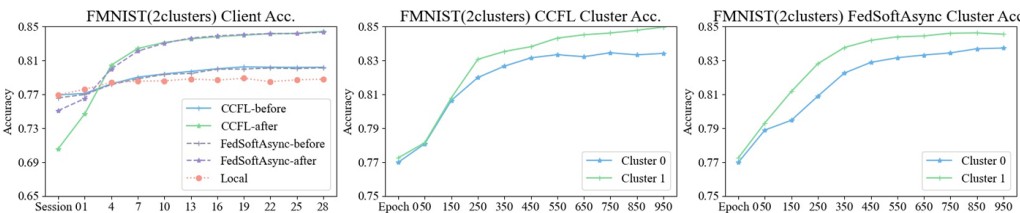

Figure 19: Timeflow accuracy of clients and clusters on FashionMNIST(2clusters) without change of distribution. The clients' distribution stay the same from beginning to end. Average accuracy of clients is shown for equal times of upload-download cycles.

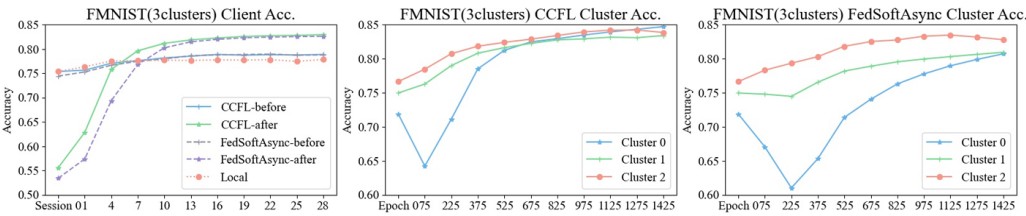

Figure 20: Timeflow accuracy of clients and clusters on FashionMNIST(3clusters) without change of distribution. The clients' distribution stay the same from beginning to end. Average accuracy of clients is shown for equal times of upload-download cycles.

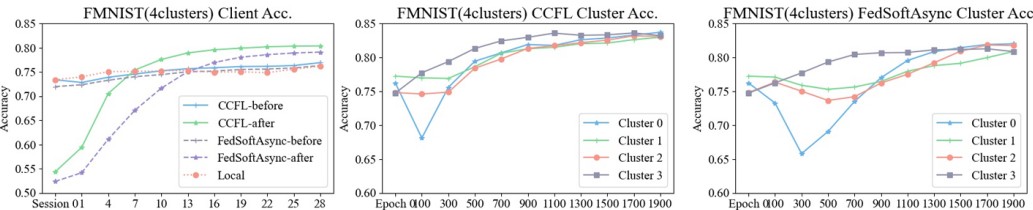

Figure 21: Timeflow accuracy of clients and clusters on FashionMNIST(4clusters) without change of distribution. The clients' distribution stay the same from beginning to end. Average accuracy of clients is shown for equal times of upload-download cycles.

Figure 18 illustrates varying sizes of the public dataset per cluster on the server (1000/2000/4000 data samples). This experiment is done by 4 clusters and 100 clients undergoing 2000 global epochs. The figure demonstrates a slight reduction in both client and cluster accuracy with a relatively smaller size of the public dataset (size of 1000). However, there's minimal disparity in the overall accuracy behavior between a dataset size of 2000 and 4000." Figure 19, 20 and 21 depict scenarios in FashionMNIST (2 clusters), FashionMNIST (3 clusters), and FashionMNIST (4 clusters) where clients' distributions remain consistent from start to finish. These experiments are done by 20 times the number of the clients of that of clusters. Each client averagely undergoes 20 upload-download cycles. These figures illustrate that our proposed method, CCFL, exhibits higher accuracy for both clusters and clients compared to the baseline methods.

