# OpenReview forum: "Client-centric Federated Learning"
_ICLR.cc/2024/Conference — Submitted to ICLR 2024_

### Official Review · Reviewer_nNv3 · 2023-11-01

**Soundness:** 2 fair
**Presentation:** 3 good
**Contribution:** 2 fair
**Rating:** 5
**Confidence:** 4

**Summary:**

The paper proposes a client-centric FL framework, where the clients have more autonomy.

**Strengths:**

- The paper is largely well-written (except some math, see below)
- The main sections 4.1 and 4.2 are well-explained, especially the use of Fig 2 helps.
- The experiment section is comprehensive.

**Weaknesses:**

Minor Issues
- $t_k$ should be defined the first time it is used in "Distributed Estimation" rather than later in "Clusters Updating''
- In Algo 2: Server Update subroutine, should the first condition be $t-\tau > \tau_0$ instead? Client Thread: where is local timestamp $t_m$ updated before sending to the server?
- For the weights $u_{mk}^t$ to sum to 1 in Algo 1, we need $c_1, c_2$ to satisfy $c_1 \sum_i \ell_i + c_2 \sum_i d_{1i} + (1-c_1-c_2) \sum_i d_{2i} = K$. Is it necessarily true that such $c_1,c_2$ will exist?

I have some doubts about the convergence analysis and experiments, see below.

Conclusion:
- What does "fluid dynamics" mean? And fluid dynamics is a completely different field in itself, so I would suggest rephrasing.

**Questions:**

My main concern with the work is its practical relevance. The authors claim to put clients at the forefront. But, if the server estimates the distribution of the clients, this is a violation of the client data privacy? How can this be reconciled since privacy is one of the privacy motivations for doing FL in the first place?

Section 1
- Among the 3 properties to encourage client participation, autonomy is well taken. But, lightweight is doubtful - the same comm/computation is needed in FedAvg. Regarding performance enhancement, one could argue that personalized FL can help with that. Please comment.

Section 4.3: several questions about the math here
- Assum. 1: What is $F$? Do you mean $F_k$ for all $k$?
- Assum. 3: what is $f_m^t$? Do you mean the first term in $h_m^t$ in (2)?
- Assum. 4: How can $w_k^*$ be unique if $F_k$ is just assumed smooth (assuming Assum. 1 is about $F_k$)? For that, we need, e.g., strong convexity. But, in Assumption 3, $f_m^t, h_m^t$, both of which depend on loss $l(\cdot)$ have bounded gradients. This is confusing.
- Theorem 1: the authors have not commented on the result. Why is the constant $\epsilon$ needed? Why does the convergence for any $k$ depend on $\sum_i a_i$? Most importantly, can you show explicitly for what choice of the learning rate $\gamma$ can we get convergence, and how do $S_k$ and $H$ affect it?

Experiments:
- Table 1: FedSoft-Async ACC for FMNIST(6). How come cluster accuracy is worse than everything else, even client bfr and local ACC.?
- Section 5.3: since the clients have the 500-2000 data points no matter how many clients are present, and the staleness bound increases with number of client, is the minimal variation with different client counts showing robustness, or an artifact of the experiment design?
- I didn't understand the reason given for worse before client accuracy with small $\rho$. If cluster models are similar, shouldn't it lead to better client accuracy as well?
- The global adjustments are claimed to mitigate adversarial behavior. But, if the server has data from all distributions, won't it recognize adversarial behavior anyway?

---

> ### Author Response · Authors · 2023-11-18
> **Response to Reviewer nNv3 (Part 1)**
>
> Thank you for highlighting the errors and concerns. We've revised the introduction to better explain the motivation behind our system's design within clustered FL and how client-centricity operates within our algorithm. In the theorem section, we clarified the roles of hyperparameters and provided detailed discussions. Additionally, we've adjusted certain term usages in the ablation study to address your concerns. **All the revised sections have been highlighted in red for your reference.**
>
> > Weakness 1: $t_k$ should be defined the first time it is used in "Distributed Estimation" rather than later in "Clusters Updating".
>
> Answer to weakness 1: Thank you for identifying this issue. To rectify this, we have made the necessary adjustments accordingly in the section titled "Distributed Estimation" (highlighted in red).
>
> > Weakness 2: In Algo 2: Server Update subroutine, should the first condition be $t-\tau > \tau_0$ instead? Client Thread: where is local timestamp $t_m$ updated before sending to the server?
>
> Response to weakness 2: Thank you for spotting this error; it should indeed be $t-\tau > \tau_0$. We have corrected this error in the paper (highlighted in red). Additionally, concerning the update of the local timestamp $t_m$, please refer to the last line of the `Client()` thread. It gets updated when the client receives the most recent timestamp from the server.
>
> > Weakness 3: For the weights $u_{mk}^t$ to sum to 1 in Algo 1, we need $c_1, c_2$ to satisfy $c_1\sum_i l_i+c_2\sum_i d_{1_i}+(1-c_1-c_2)\sum_id_{2i} = K$. Is it necessarily true that such $c_1, c_2$ will exist?
>
> Answer to weakness 3: Yes, the way the distribution $u_{mk}^t$ is computed is designed such that $\sum_k u_{mk}^t =1 $ holds for arbitrary values of $c_1, c_2$ such that $c_1 \in [0, 1]$, $c_2 \in [0, 1]$ and $c_1+c_2 \in [0, 1]$.
>
> > Weakness 4: What does "fluid dynamics" mean? And fluid dynamics is a completely different field in itself, so I would suggest rephrasing.
>
> Answer to weakness 4: We apologize for any potential confusion arising from our terminology. By 'dynamics', we specifically refer to the changes in clients' data distributions. To avoid any misinterpretation, we have now removed the term 'fluid dynamics' from the conclusion section.
>
> > Main concern: My main concern with the work is its practical relevance. The authors claim to put clients at the forefront. But, if the server estimates the distribution of the clients, this is a violation of the client data privacy? How can this be reconciled since privacy is one of the privacy motivations for doing FL in the first place?
>
> Answer to main concern: Thank you for raising this insightful question. We want to clarify that the focus of client-centrism considered in this paper is to enhance the performance and efficiency of clients in FL, but not particularly on privacy. Nevertheless, we emphasize that the potential information leakage in our framework remains the same with that of a canonical FL framework (e.g., FedAvg). Specifically, in our design, a client uploads its local model trained on private data to the server and nothing more, which is identical to what is done in canonical FL frameworks. Therefore, our design does not amplify the risk of privacy violation, and any information about private data that can be inferred in our framework can also be inferred by curious server in canonical FL. We acknowledge that privacy is certainly a critical aspect in FL, and how to further enhance clients' data privacy in our client-centric framework is an interesting and important future research question.

---

> ### Author Response · Authors · 2023-11-18
> **Response to Reviewer nNv3 (Part 2)**
>
> > Question 1: Among the 3 properties to encourage client participation, autonomy is well taken. But, lightweight is doubtful - the same comm/computation is needed in FedAvg. Regarding performance enhancement, one could argue that personalized FL can help with that. Please comment.
>
> Answer to question 1: We deeply regret any confusion that may have arisen from our presentation. To clarify, our comparison focused on assessing the communication and computation overheads within the domain of clustered FL methods, rather than the general FedAvg setting. In our scenario, where clients possess a mixture of distributions subject to change, we employ clustered FL to address this issue. However, existing clustered FL frameworks place communication and computation burdens on the clients, diverging from our client-centric philosophy, which aims to deliver optimal performance with minimal computational and communication resources. To align with this objective, we developed a novel clustered FL framework that offloads all communication and computation tasks to the server, hence our reference to it as 'lightweight'. Regarding the performance enhancement, we referred to the performance gain of the client through model updating with the server, who possesses cluster models trained over more data from other clients. This performance gain encourages clients to join the proposed CCFL framework. As pointed out by the reviewer, our approach indeed falls into the personalized FL category, as each time a client requests a model update with the server, the server returns it a personalized model according to estimated cluster distribution.  We have now modified introduction to make this point more clear (modifications highlighted in red).
>
> > Question 2: What is $F$? Do you mean $F_k$ for all $k$?
>
> Answer to question 2: Yes. To make it more clear, we have now restated Assumption 1 (highlighted in red).
>
> > Question 3: what is $f_m^t$? Do you mean the first term in $h_m^t$ in (2)?
>
> Response to question 3: Yes. To make it more clear, we have now restated Assumption 3 (highlighted in red).
>
> > Question 4: How can $w_k^*$ be unique if $F_k$ is just assumed smooth (assuming Assum. 1 is about $F_k$)? For that, we need, e.g., strong convexity. But, in Assumption 3, $f_m^t$, $h_m^t$, both of which depend on loss  have bounded gradients. This is confusing.
>
> Response to question 4: Thank you for highlighting the missing requirement of convexity. Convexity is indeed essential in our proof, and we have included it in the restated Assumption 1 (highlighted in red). We apologize for any confusion this may have caused.

---

> ### Author Response · Authors · 2023-11-18
> **Response to Reviewer nNv3 (Part 3)**
>
> > Question 5: the authors have not commented on the result. Why is the constant $\epsilon$ needed? Why does the convergence for any k depend on $\sum_i{a_i}$? Most importantly, can you show explicitly for what choice of the learning rate $\gamma$ can we get convergence, and how do $S_k$ and $H$ affect it?
>
> Answer to question 5: We apologize for the oversight in not discussing the final convergence theorem. We have now included additional discussions on our convergence analysis (highlighted in red) for your reference.
> * We need the small constant $\epsilon$ to further upper bound the right hand side of inequality (7), where we need a lower bound of $\mathbb{E}[\langle \nabla H_k(w_{h-1}; \mathbf{w}_\tau), \nabla h_m(w _{h-1}; \mathbf{w} _\tau)\rangle]$. We can show that there exists a positive constant $\epsilon$ such that for $\rho \geq  \frac{2V_1+\frac{1}{2}\left\|w _{h-1}-\mathbf{w} _\tau\right\|^2+\sqrt{4\left\|w _{h-1}-\mathbf{w} _\tau\right\|^2(1+V_1)\epsilon}}{2\left\|w _{h-1}-\mathbf{w} _\tau\right\|^2}$ and all possible $w _{h-1}, \mathbf{w} _\tau$, $\mathbb{E}[\langle \nabla H_k(w _{h-1}; \mathbf{w} _\tau), \nabla h_m(w _{h-1}; \mathbf{w} _\tau)\rangle]$ can be lower bounded by $\epsilon\left\|\nabla F_k(w _{h-1})\right\|^2$, then we can upper bound the right hand side of inequality (7) by $\epsilon\left\|\nabla F _k(w _{h-1})\right\|^2$.
> * To the question why the convergence for any k depends on $\sum_i{a_i}$, it is because the convergence bound of model $k$ contains a term $\left\| \mathbf{w} _\tau - \mathbf{w} _{s-1}\right\|$ , which is shown in (13)-(18) to be upper bounded by $\left(2\sum _{i=1}^K a  _i+(2K+1)a _k+K\right)\Delta$. Intuitively speaking, the model $\mathbf{w} _\tau$ is aggregated by all cluster models $\mathbf{w} _1, ..., \mathbf{w} _K$, so $\left\|\mathbf{w}  _\tau-\mathbf{w}  _{s-1}\right\|$ can be seen as the distance from an aggregation of all cluster models at epoch $\tau$, namely the aggregation of $\mathbf{w} _i^\tau, i=1, ... K$, to the cluster $k$'s last model $w _{s-1}$. Since we assume for $\forall i \in [K]$, $\left\|\mathbf{w} _i\right\| \leq a _i \Delta$, thus $a _i$ can help bound the distance from cluster $k$'s model to other clusters' model $\mathbf{w} _i$. As a result, $\sum _i a _i$ is needed here to bound the $l _2$-norm distance between $\mathbf{w} _\tau$ and cluster $k$'s last model $\mathbf{w} _{s-1}$, namely $\left\|\mathbf{w} _\tau-\mathbf{w} _{s-1}\right\|$, and finally contributes to the upper bound of the convergence of distribution $k$.
> * For the choice of $\gamma, S _k, H _{min}$ and $H _{max}$, as shown in the proof in Appendix A.2, the upper bound of $\mathbb{E}[\left\|\nabla F _k(w)\right\|^2]$ is proven to be $\frac{\mathbb{E}[F _k(\mathbf{w} _{0})-F _k(\mathbf{w} _{S _k})]}{\alpha _0\gamma\epsilon S _k H _{min}}+\frac{\left(\frac{L _k}{2}+\rho H _{max}+\frac{\rho H _{max}^2}{2}\right)\gamma H _{max}V_2}{\epsilon H _{min}} +\frac{\sqrt{V _1} \left(2\sum _{i=1}^K a _i+(2K+1)a _k+K\right)\Delta}{\gamma\epsilon H _{min}}+\frac{\left(\frac{L _k}{2}+\rho\right)\left(2\sum _{i=1}^K a _i+(2K+1)a _k+K\right)^2\Delta^2}{\gamma\epsilon H _{min}}$. Here we do not specify particular values for variables $H _{min}$, $H _{max}$, $S _k$, and $\gamma$. One possible choice of these variables, as suggested in [1], is that $\gamma = \frac{1}{\sqrt{S _k}}$ and $S _k = H _{min}^5$, leading to an upper bound of $\mathcal{O}\left(\frac{1}{\epsilon H^3 _{min}}+\frac{\delta^3}{\epsilon\sqrt{H _{min}}}+\frac{K\delta}{\epsilon\sqrt{H _{min}}}+\frac{K^2\delta^2\Delta}{\epsilon\sqrt{H^5 _{min}}}+\frac{K^2\delta^2\Delta^2}{\epsilon\sqrt{H^3 {min}}}\right)$ with $\delta = \frac{H _{max}}{H _{min}}$. Generally speaking, the upper bound increases with $H _{max}$, and decreases with $S _k$ and $H _{min}$. Intuitively, $S _k$ is the iteration counts of learning cluster $k$, thus more iterations at the server lead to a smaller upper bound of $\mathbb{E}[\left\|\nabla F _k(w)\right\|^2]$. $H _{min}$, $H _{max}$ are respectively the least and most local iteration counts before a single client updates to the server in one cycle, then bigger $H _{min}$ indicates a more sufficient local training, and smaller $H _{max}$ reduces the influence of overfitting.
> *  In conclusion, this theorem indicates that if a client's model $w$ undergoes continuous training on data from distribution $k$, meaning that a portion of the client's data consistently originates from distribution $k$, then the $l_2$-norm of the gradient of the model loss on cluster $k$ will converge to a specific point (always less than $\infty$). For any data distribution $k$ continuously sampled by a client, the proposed algorithm guarantees the client's model to have good performance on this particular distribution $k$.
>
> [1] Xie, C., Koyejo, S., Gupta, I., 2019. Asynchronous federated optimization. arXiv preprint
> arXiv:1903.03934 .

---

> ### Author Response · Authors · 2023-11-18
> **Response to Reviewer nNv3 (Part 4)**
>
> > Question 6: Table 1: FedSoft-Async ACC for FMNIST(6). How come cluster accuracy is worse than everything else, even client bfr and local ACC.?
>
> Response to question 6: Thank you for your meticulous observation. To address the query regarding accuracy, please refer to Figure 9 in Appendix A.3, titled "Timeflow accuracy of clients and clusters on FashionMNIST (6 clusters)". As depicted in the figure, lower accuracy for cluster accuracy arises from certain significantly low-performing clusters, such as cluster 1. This occurrence can be attributed to the limitations of `FedSoft` in accurately estimating distributions, particularly when dealing with a relatively large number of clusters. Previous experiments, as detailed in [1], primarily focused on 2 or 3 clusters. However, in the case of 6 clusters, the distribution estimation by `FedSoft` was notably inaccurate, sometimes approximating an equal distribution of 1/6 for each cluster. Consequently, this inaccurate estimation led to clusters being updated by models that did not align with their corresponding distributions, thereby resulting in lower cluster accuracy over time.
>
> [1] Ruan, Y., Joe-Wong, C., 2022. Fedsoft: Soft clustered federated learning with proximal local updating, in: Proceedings of the AAAI Conference on Artificial Intelligence, pp. 8124–8131.
>
> > Question 7: Section 5.3: since the clients have the 500-2000 data points no matter how many clients are present, and the staleness bound increases with number of client, is the minimal variation with different client counts showing robustness, or an artifact of the experiment design?
>
> Response to question 7: Thank you for this insightful question. We conducted multiple experiments varying the number of clients, and the consistent results across different client counts suggest that the variation in accuracy is not coincidental. One possible explanation could be the relatively small size of our datasets, namely FashionMNIST and CIFAR100, both containing 60,000 data samples. With datasets comprising millions or billions of samples, a larger number of clients could potentially result in a broader range of data samples, leading to improved results. Another plausible reason could stem from the nature of the models, such as CNN and ResNet, and the simplicity of datasets like FashionMNIST and CIFAR100. These models and datasets may require only a limited amount of training data to achieve satisfactory performance, thus minimizing the impact of varying client counts on the overall outcome.
>
> > Question 8: I didn't understand the reason given for worse before client accuracy with small $\rho$. If cluster models are similar, shouldn't it lead to better client accuracy as well?
>
> Response to question 8: Thank you for your intriguing question. It's important to note that the "before client accuracy" does not inherently correlate with the accuracy of cluster models, particularly in cases where a small scaling term, $\rho$, is applied. With a smaller scaling term, locally trained client models have a tendency to overfit on their respective local data. Consequently, the integration of cluster models in such scenarios helps to achieve a substantial improvement in "after client accuracy" compared to the "before client accuracy".
>
> > Question 9: The global adjustments are claimed to mitigate adversarial behavior. But, if the server has data from all distributions, won't it recognize adversarial behavior anyway?
>
> Answer 9: We sincerely apologize for any confusion caused by the use of the term "adversarial". It's essential to clarify that in our framework, we assume all clients to be honest. The global adjustments we introduce serve to *"filter out"* models that might have a detrimental impact on the global cluster models. Due to the non-iid nature of the clients' data, certain trained models may not enhance the overall model performance and could, in some cases, even reduce global model accuracy. In such instances, the global adjustments effectively prevent these potentially harmful models from impacting the server cluster accuracy. We have rectified this improper use of the term "adversarial behavior" in the relevant section of our paper (highlighted in red) for your reference.

---

> ### Author Response · Authors · 2023-11-20
> **Remind for feedback on our rebuttal**
>
> Dear Reviewer nNv3,
>
> We are approaching the end of the discussion phase, and we have unfortunately received no feedback from you on our rebuttal.
>
> Please can we kindly ask you to take a look at our responses, and let us know whether we have clarified your questions and addressed your concerns?
>
> Specifically, We've revised the introduction to better explain the motivation behind our system's design within clustered FL and how client-centricity operates within our algorithm. In the theorem section, we clarified the roles of hyperparameters and provided detailed discussions. Additionally, we've adjusted certain term usages in the ablation study to address your concerns. All the revised sections have been highlighted in red for your reference.
>
> Thank you very much again for the time you spent reviewing.
>
> Paper3372 Authors

---

### Official Review · Reviewer_aGfK · 2023-11-03

**Soundness:** 3 good
**Presentation:** 3 good
**Contribution:** 2 fair
**Rating:** 5
**Confidence:** 3

**Summary:**

The paper presents a client-centric framework for federated learning, where a client can decide when to participate in the collaboration and  where the result of an update is a model tailored to the local task. To accomplish this, the server maintains several cluster models, estimates a client's distribution based on available public data, and returns a personalized model based on this distribution. Results are demonstrated on three standard machine learning tasks.

**Strengths:**

- The philosophical motivation presented by the manuscript is quite compelling: a client-centric federated framework.
- Handling local client distributions that change over time is an important practical scenario.
- The paper is overall well written and well presented.

**Weaknesses:**

- Although the paper provides a convergence result in Section 4.3, there is no discussion about it. Does this result elucidate anything about the behavior of the algorithm under the given assumptions? I admit I did not check the math carefully.
- The presented experiments don't account for variance across repetitions. None of the plots have error bars, and the error bars in Table 1 represent variance across clients in one single repetition. This would make the results more robust, particularly when the results are close (e.g., in Table 1, when the number of clusters is 4).
- There is no discussion on how to choose the number of clusters or where the public cluster data would come from.

**Questions:**

- I enjoy the client-centric philosophy presented by the paper, but I am not sure if the method truly adheres to this philosophy. I understand this client-centrism is exemplified by the asynchronous nature of the framework, giving clients autonomy over when to send updates to the server. Similarly, the philosophy is reflected by the fact that the updates result in a personalized model for the clients. However, I struggle to reconcile the idea of a client-centric framework with the fact that the server is still a key element and bottleneck in this algorithm. In fact, the paper recognizes that more tasks have been placed on the server compared to previous work (e.g., distribution estimation). This is the main reason for my rating, as I believe the manuscript should either be more careful in addressing this tension, or be more specific in their title/motivation.
- In the first paragraph of Section 4.2, there is a reference to "aggregation of global models". I assume this refers to the cluster models? This particular term was confusing when I first read the manuscript.
- When explaining the experimental set-up in Section 5.1, there is no detailed description of how the clients' data changes over time. If this is in the appendix, please include a reference.
- Just to clarify: in Figure 4(a), you know the true distribution of the clients because you track how their data changes through time?

---

> ### Author Response · Authors · 2023-11-18
> **Response to Reviewer aGfK (Part 1)**
>
> Thank you for the valuable feedback. We've thoroughly discussed Theorem 1 in Section 4.3, highlighting its impact on the algorithm's behavior under specific assumptions. Error bars representing result variance across repetitions have been added to figures, enhancing result robustness.In the introduction part, we have revised our paper to address your worries why server stands as a key element in our system. Shortly speaking, the server's role is to support clients by taking on the computation and communication burden, aligning with our goal of efficient and lightweight client operations. **The revised sections are highlighted in red for your convenience.**
>
> > Weakness 1: Although the paper provides a convergence result in Section 4.3, there is no discussion about it. Does this result elucidate anything about the behavior of the algorithm under the given assumptions? I admit I did not check the math carefully.
>
> Response to weakness 1: Thank you for highlighting this concern. We have now added discussions regarding Theorem 1's implication after the theorem statement (indicated in red). Specifically, the theorem indicates that if a client's model $w$ undergoes continuous training on data from distribution $k$, meaning that a portion of the client's data consistently originates from distribution $k$, then the $l_2$-norm of the gradient of the model loss on cluster $k$ will converge to a specific point (always less than $\infty$). For any data distribution $k$ continuously sampled by a client, the proposed algorithm guarantees the client's model to have good performance on this particular distribution $k$.
>
> > Weakness 2: The presented experiments don't account for variance across repetitions. None of the plots have error bars, and the error bars in Table 1 represent variance across clients in one single repetition. This would make the results more robust, particularly when the results are close (e.g., in Table 1, when the number of clusters is 4).
>
> Response to weakness 2: Thank you for highlighting the absence of error bars, which could potentially affect the robustness of our experiments. We have incorporated error bars, indicated by shaded areas, based on three repetitions for each experiment in Figure 3 in the main body of the paper, Figures 6-17 in Appendix A.3. Table 1 is also updated with average accuracy across 3 trials.
>
> > Weakness 3: There is no discussion on how to choose the number of clusters or where the public cluster data would come from.
>
> Response to weakness 3: Thank you for highlighting the missing information. The total number of clusters, denoted as $K$, should indeed be considered prior knowledge, which the service provider may decide based on the type of service (e.g., skincare maintenance), or deduce from its available data, possibly encompassing various user attributes such as genders or ethnicities. The public cluster data can be collected initially by the service provider, and only small amount can help to achieve high cluster and client performance: see experiment on different sizes of public datasets in Figure 18 in Appendix A.4. With only 1000 data points at the server, a client accuracy of 80\% and a cluster accuracy of 82\% can be achieved for FashionMNIST dataset. We have also provided further clarification on this aspect in our problem definition (Section 3), highlighted in red for your convenience.

---

> ### Author Response · Authors · 2023-11-18
> **Response to Reviewer aGfK (Part 2)**
>
> > Question 1: I enjoy the client-centric philosophy presented by the paper, but I am not sure if the method truly adheres to this philosophy. I understand this client-centrism is exemplified by the asynchronous nature of the framework, giving clients autonomy over when to send updates to the server. Similarly, the philosophy is reflected by the fact that the updates result in a personalized model for the clients. However, I struggle to reconcile the idea of a client-centric framework with the fact that the server is still a key element and bottleneck in this algorithm. In fact, the paper recognizes that more tasks have been placed on the server compared to previous work (e.g., distribution estimation). This is the main reason for my rating, as I believe the manuscript should either be more careful in addressing this tension, or be more specific in their title/motivation.
>
> Answer to Question 1: We sincerely regret any confusion and misunderstanding that may have arisen. To establish a client-centric service, we have envisioned the server's role as an assistant to the clients. In addressing our problem setting where clients' data exhibit a mixture of distributions that dynamically change over time, we opted for a clustered FL framework to resolve this challenge. It's noteworthy that existing clustered FL methodologies burden clients with all computation and communication loads, which doesn't align with the client-centric paradigm we strive for. Specifically, **our notion of client-centrism is for the client to achieve high performance with minimum computation and communication complexities**. Therefore, we restructured the clustered FL system, moving the major computation burden of distribution estimation onto the server, without requiring communicating all cluster models to the client, thereby significantly reducing communication and computation loads on clients. This redesign adheres to our goal of performant and lightweight clients. Importantly, a client-centric approach doesn't entail transferring all computation to clients but rather ensures they attain optimal performance with minimal resource utilization. To clarify this point, we have now revised portions of our introduction (highlighted in red) to better explain this motivation.
>
>
> > Question 2: In the first paragraph of Section 4.2, there is a reference to "aggregation of global models". I assume this refers to the cluster models? This particular term was confusing when I first read the manuscript.
>
> Response to question 2: We sincerely apologize for this confusion. To clarify, yes, our reference to "aggregation of global models" indeed refers to the process of combining cluster models. We have revised the first paragraph of Section 4.2 (highlighted in red) to make this clear.
>
> > Question 3: When explaining the experimental set-up in Section 5.1, there is no detailed description of how the clients' data changes over time. If this is in the appendix, please include a reference.
>
> Answer to question 3 : Thanks for your careful reading of the our experiments. In response to your feedback, we have now included in Appendix A.1 (highlighted in red) detailed descriptions on how clients' data change over time in our experiments.
>
> > Question 4: Just to clarify: in Figure 4(a), you know the true distribution of the clients because you track how their data changes through time?
>
> Answer to question 4: Yes, we monitored the evolution of clients' data throughout our experiment to facilitate a comparative analysis between our estimation method and the baseline.

---

> ### Author Response · Authors · 2023-11-20
> **Remind for feedback on our rebuttal**
>
> Dear Reviewer aGfK,
>
> We are approaching the end of the discussion phase, and we have unfortunately received no feedback from you on our rebuttal.
>
> Please can we kindly ask you to take a look at our responses, and let us know whether we have clarified your questions and addressed your concerns?
>
> Specifically, We've thoroughly discussed Theorem 1 in Section 4.3, highlighting its impact on the algorithm's behavior under specific assumptions. Error bars representing result variance across repetitions have been added to figures, enhancing result robustness.In the introduction part, we have revised our paper to address your worries why server stands as a key element in our system. Shortly speaking, the server's role is to support clients by taking on the computation and communication burden, aligning with our goal of efficient and lightweight client operations. The revised sections are highlighted in red for your convenience.
>
> Thank you very much again for the time you spent reviewing.
>
> Paper3372 Authors

---

### Official Review · Reviewer_7mm9 · 2023-11-04

**Soundness:** 3 good
**Presentation:** 2 fair
**Contribution:** 2 fair
**Rating:** 5
**Confidence:** 3

**Summary:**

The authors propose a new framework in FL where clients determine asynchronous when to update a local trained model to a server. The framework focuses on clustered FL and local distribution shifts in the clients and effects of staleness. Results are presented in comparison ton an Async-FL algorithm on FashionMNIST, CIFAR-100, MiniImagenet

**Strengths:**

- The idea to have clients control the update process is interesting and novel
- The solution is sensible and the results are promising

**Weaknesses:**

- The technical contributions can be difficult to follow, although the paper seems to suggest their main contribution is to have a client centric update process a lot of the focus is on clustered FL and client level distribution shifts making the exact setting and problem a bit contrived. It would be good to isolate these factors better both in the motivation of the method and in the experiments
- How does client level distribution shift relate to the area of continual federated learning
- It is difficult to asses if the compared baseline is the most appropriate, the baseline methods should be better motivated
- There are a lot of hyperparameters, how would these be decided in practice

**Questions:**

How would this framework perform in a setting without clusters or assumptions of local distribution shifts

---

> ### Author Response · Authors · 2023-11-18
> **Response to Reviewer 7mm9 (Part 1)**
>
> Thank you sincerely for your insightful comments and suggestions. We have revised our paper to enhance the clarity surrounding the interconnections among client-centric FL, clustered FL, and distribution shifts, particularly in the introduction section. To be more specific, we have provided a clear distinction between our setting and Federated Continual Learning (FCL), provided a detailed rationale for selecting the baseline method, and extensively elaborated on hyperparameters and ablation studies. Additionally, we've included diverse scenarios you mentioned in the appendix through added experiments. All revised sections are **highlighted in red** for your convenience and reference in the paper.
>
> > Weakness 1: The technical contributions can be difficult to follow, although the paper seems to suggest their main contribution is to have a client centric update process a lot of the focus is on clustered FL and client level distribution shifts making the exact setting and problem a bit contrived. It would be good to isolate these factors better both in the motivation of the method and in the experiments.
>
> Response to weakness 1: Thank you for your valuable suggestion regarding the isolation of these factors. We have revised the introduction section (highlighted in red) to clearly describe the relationship between client-centric FL, clustered FL, and distribution shifts. Specifically, we now clearly describe the setting of the considered client-centric FL: a client's local data is sampled from a mixture of distributions whose mixing ratio may change over time, hence the client may need to update its model with the server to accommodate the new distribution. We also provide real-life examples to illustrate the significance of the considered client-centric FL problem. On the other hand, as now clearly stated in the introduction, clustered FL, in which a group of $K$ cluster models (one for each distribution) are hosted at the server, and are updated using the uploaded data from clients, is adopted in this paper as a basic framework to tackle the challenges in the considered client-centric FL problem. Our main contribution of this paper is to formulate the client-centric FL problem, and propose a clustered FL based protocol for this problem (where when to do model update is autonomously decided by each individual client) with high cluster and client accuracies, and minimum communication and computation costs at clients.
>
> > Weakness 2: How does client level distribution shift relate to the area of continual federated learning?
>
> Answer to weakness 2: We regret any confusion that may have arisen regarding the relationship between distribution shift in our setting and Federated Continual Learning (FCL). It's critical to note that our setting is drastically distinct from FCL. FCL primarily focuses on mitigating catastrophic forgetting induced by arrivals of new tasks at each client, striving to achieve proficient performance across both past and current tasks. Conversely, our focus lies solely on maintaining high performance on the current task. We aim to provide clients with models better suited to their update-to-date data when their data distribution undergoes a shift. In order to avoid any confusion, we have revised the related work section to explicitly compare our setting with FCL (highlighted in red).
>
> > Weakness 3: It is difficult to asses if the compared baseline is the most appropriate, the baseline methods should be better motivated.
>
> Answer to weakness 3: Thank you for your valuable suggestion. To clarify, to the best of our knowledge, there hasn't been any prior research conducted in the context of combining asynchrony and soft-clustering in FL, which is the setting considered in our client-centric framework. Consequently, for comparison, we adapt the state-of-the-art soft-clustering FL framework, `FedSoft`, initially designed for synchronous FL, to an asynchronous setting to create our comparison baseline `FedSoft-Async`. To better clarify the motivation behind choosing this baseline method, we have now incorporated a detailed explanation of this rationale into the experimental setup section (highlighted in red).
>
> > Weakness 4: There are a lot of hyperparameters, how would these be decided in practice?
>
> Response to weakness 4: Thank you for your inquiry. For comprehensive insights into our experiments, we have meticulously documented all the specific hyperparameters utilized in our study in Table 2 in Appendix A.1. Additionally, we have conducted ablation studies involving variations on the number of clients, different regularization terms represented by $\rho$, and various adjustment thresholds in Section 5.3 Ablation study and Appendix A.4. If you require more detailed information about particular hyperparameters and their impact on the algorithm's performance, please feel free to specify your query. We would be more than willing to provide further clarification and elaboration on these aspects.

---

> ### Author Response · Authors · 2023-11-18
> **Response to Reviewer 7mm9 (Part 2)**
>
> > Question 1: How would this framework perform in a setting without clusters or assumptions of local distribution shifts?
>
> Response: Thank you for the question. Our framework is designed as a comprehensive solution, encompassing the mentioned cases as special instances. We have also included experiments on FashionMNIST (1 cluster) in the Appendix A.3, as depicted in Figure 6, and CIFAR-100 (1 cluster), depicted in Figure 10. In these two experiments,  the local distribution would shift after the clients update their local model. What's more, we included experiments on FashionMNIST where clients' data distribution stay the same as shown in Appendix A.4, Figure 19, 20, 21. This additional experiment illustrates that our method consistently outperforms `FedSoft-Async` concerning cluster and client accuracy.

---

> ### Author Response · Authors · 2023-11-20
> **Remind for feedback on our rebuttal**
>
> Dear Reviewer 7mm9,
>
> We are approaching the end of the discussion phase, and we have unfortunately received no feedback from you on our rebuttal.
>
> Please can we kindly ask you to take a look at our responses, and let us know whether we have clarified your questions and addressed your concerns?
>
> Specifically, we have revised our paper to enhance the clarity surrounding the interconnections among client-centric FL, clustered FL, and distribution shifts, particularly in the introduction section. To be more specific, we have provided a clear distinction between our setting and Federated Continual Learning (FCL), provided a detailed rationale for selecting the baseline method, and extensively elaborated on hyperparameters and ablation studies. Additionally, we've included diverse scenarios you mentioned in the appendix through added experiments. All revised sections are highlighted in red for your convenience and reference in the paper.
>
> Thank you very much again for the time you spent reviewing.
>
> Paper3372 Authors

---

### Author Response · Authors · 2023-11-18
**General Response**

We are very thankful for the area chair to coordinate the review of our manuscript and grateful to the reviews for their valuable feedback. We very much appreciate the assessment of work as **novel (Reviewer 7mm9), interesting (Reviewer 7mm9), compelling (Reviewer aGfK)** and **well written (Reviewer aGfK, nNv3)**.

The authors have provided point-to-point responses to the comments raised by the reviewers. A summary of them is as follows:
* In response to concerns raised by **Reviewers 7mm9, aGfK, and nNv3** regarding the **motivation** behind client-centric FL and clarifying the relationships between clients, server, clusters, and distribution drift, the authors have meticulously rewritten the introduction (Section 1) of the paper (highlighted in red). This revision emphasizes our approach to addressing the challenge of dynamically changing data distributions in clients' datasets using a clustered FL framework. Our focus on client-centricity aims for optimal client performance with minimum computational and communication loads.
* To address concerns raised by **Reviewers aGfK and nNv3** regarding the **convergence analysis** and its impact, the authors have enhanced Section 4.3. This section now provides a more detailed explanation on the implication of our main theorem. Generally speaking, for any data distribution $k$ continuously sampled by a client, the proposed algorithm guarantees the client’s model to have good performance on this particular distribution $k$ by giving an upper bound of the expectation of the gradient of the loss function of client model.
* In response to **Reviewer 7mm9**'s suggestion, **additional ablation studies** have been conducted where the server holds only one cluster or clients' distributions remain unchanged. Furthermore, as recommended by **Reviewer aGfK**, **error bars** representing variances for 3 repetitive experiments have been incorporated into Figures 3 and 6-17. Table 1 is updated with average accuracy across 3 trials.
* Vague concepts highlighted by the reviewers have been addressed: in Section 2 (Related Works) to elucidate the relationship between FCL and our proposed algorithm as suggested by **Reviewer 7mm9**; details on determining the number of clusters and public cluster data in Section 3 as per **Reviewer aGfK**'s suggestion; a comprehensive description of the experiment setting in Appendix A.1 following **Reviewer aGfK**'s advice; and adjustments in Section 5.3 (Ablation Study) based on **Reviewer nNv3**'s suggestions.

**The authors would like to kindly remind the reviewers to take a look at the responses and see whether the raised concerns have been well addressed. Thank you for your help and expertise. We look forward to hearing from you again.**

---

### Meta-Review · Area_Chair_FTgz · 2023-12-10

**Metareview:**

This work proposes a novel federated learning approach called Client-centric Federated Learning. In CCFL, the main idea is that, rather than server synchronizing client updates, the authors propose a strategy where clients can upload their models to server on demand. The approach of the authors relies on retaining K models and a clustering strategy where each client's distribution at a particular epoch is softly matched to one of the K clusters. Most of the reviewers and I have found the idea, approach, and the proposed framework promising. However, there are a few bottlenecks highlighted by reviewers that resulted in a reject recommendation. A central issue is the convergence analysis. As pointed out by Reviewer nNv3, the initial version of this result was not fully rigorous missing key assumptions and confusing notation. Even the revised version of Thm 1 has a confusing notation because the subscript of $\textbf w_k$ sometimes refers to k'th cluster and sometimes to k'th epoch. Additionally, I am concerned that the analysis does not reflect some of the core benefits of clustering (e.g. aggregating similar clients) or fundamental tradeoffs associated with the proposed method. Overall, the authors can benefit from revising the main theory for clarity and rigor and incorporate reviewer suggestions.

**Justification For Why Not Higher Score:**

N/A

**Justification For Why Not Lower Score:**

N/A

---

### Decision · Program_Chairs · 2024-01-16

Reject